# Compressive stress-mediated p38 activation required for ERα + phenotype in breast cancer

Pauliina M. Munne [1], Lahja Martikainen [2,14], Iiris Räty [1,14], Kia Bertula [2], Nonappa [2,3], Janika Ruuska [1], Hanna Ala-Hongisto [1], Aino Peura [1], Babette Hollmann[1], Lilya Euro [4], Kerim Yavuz [5], Linda Patrikainen[1], Maria Salmela[1], Juho Pokki [6], Mikko Kivento [7], Juho Väänänen [7], Tomi Suomi [8], Liina Nevalaita [1], Minna Mutka [9], Panu Kovanen [9], Marjut Leidenius[10], Tuomo Meretoja[10], Katja Hukkinen[11], Outi Monni [7], Jeroen Pouwels[1], Biswajyoti Sahu [5], Johanna Mattson [12], Heikki Joensuu [12], Päivi Heikkilä[9], Laura L. Elo [8], Ciara Metcalfe[13], Melissa R. Junttila[13], Olli Ikkala [2,3] & Juha Klefström [1]✉

Breast cancer is now globally the most frequent cancer and leading cause of women's death. Two thirds of breast cancers express the luminal estrogen receptor-positive (ERα + ) phenotype that is initially responsive to antihormonal therapies, but drug resistance emerges. A major barrier to the understanding of the ERα-pathway biology and therapeutic discoveries is the restricted repertoire of luminal ERα + breast cancer models. The ERα + phenotype is not stable in cultured cells for reasons not fully understood. We examine 400 patient-derived breast epithelial and breast cancer explant cultures (PDECs) grown in various three-dimensional matrix scaffolds, finding that ERα is primarily regulated by the matrix stiffness. Matrix stiffness upregulates the ERα signaling via stress-mediated p38 activation and H3K27me3-mediated epigenetic regulation. The finding that the matrix stiffness is a central cue to the ERα phenotype reveals a mechanobiological component in breast tissue hormonal signaling and enables the development of novel therapeutic interventions. Subject terms: ER-positive (ER + ), breast cancer, ex vivo model, preclinical model, PDEC, stiffness, p38 SAPK.

[1] Finnish Cancer Institute, FICAN South Helsinki University Hospital & Translational Cancer Medicine, Medical Faculty, University of Helsinki. Cancer Cell Circuitry Laboratory, PO Box 63 Haartmaninkatu 8, 00014 University of Helsinki, Helsinki, Finland. [2] Department of Applied Physics, Molecular Materials Group, Aalto University School of Science, PO Box, 15100, FI-00076 Espoo, Finland. [3] Department of Bioproducts and Biosystems, Aalto University School of Chemical Engineering, Espoo, Finland. [4] Research Program of Stem Cells and Metabolism, Biomedicum Helsinki, University of Helsinki, 00290 Helsinki, Finland. [5] Applied Tumor Genomics Research Program, Enhancer Biology Laboratory, Faculty of Medicine, University of Helsinki, Helsinki, Finland. [6] Department of Electrical Engineering and Automation, Aalto University, Espoo, Finland. [7] Applied Tumor Genomics Research Program, Oncogenomics Laboratory, University of Helsinki, Helsinki, Finland. [8] Turku Bioscience Centre, University of Turku and Åbo Akademi University, FI-20520 Turku, Finland. [9] Department of Pathology, HUSLAB and Haartman Institute, Helsinki University Central Hospital and University of Helsinki, Helsinki, Finland. [10] Breast Surgery Unit, Helsinki University Central Hospital, Helsinki, Finland. [11] Department of Mammography, Helsinki University Central Hospital, Helsinki, Finland. [12] Department of Oncology, University of Helsinki & Helsinki University Hospital, Helsinki, Finland. [13] Genentech Inc., 1 DNA Way, South San Francisco, CA 94080, USA. [14] These authors contributed equally: Lahja Martikainen, Iiris Räty. ✉email: Juha.Klefstrom@helsinki.fi

Breast cancers are commonly divided into four molecular subtypes based on specific therapeutically actionable bio-markers and gene expression profiles[1]. About 80% of all newly diagnosed breast cancers have luminal cell phenotype and they express estrogen receptor (ERα). These breast cancers are called either luminal A or B subtype and they have relatively good prognosis compared to the more aggressive ERα-negative subtypes; HER2-enriched (HER2 + , ERα-) and triple-negative/basal-like (TNBC) breast cancer[2,3]. While the overall prognosis of localized, early-stage breast cancer is usually excellent, overtly metastatic disease is still considered incurable (www.seer.cancer.gov). The treatment options for advanced ERα + (HER2-) breast cancer commonly include endocrine therapies, such as aromatase inhibitors or selective estrogen receptor degraders/ modulators (SERDs and SERMs), administered either alone or in combination with targeted therapies like cyclin-dependent kinase inhibitors or mTOR inhibitors[4]. The commonness of ERα positive luminal breast cancer together with the effectiveness and widespread use of ERα pathway inhibitors in treatment predicts that this biology will remain the major focus of research and drug development.

Despite the need for novel ERα pathway-targeting drugs, only a few ERα + preclinical models are available for the drug discovery, development and testing. The establishment of ERα + luminal breast cancer cell lines has turned out to be a challenging task for reasons not entirely clear. In cell culture systems, the luminal ERα + tumor cells are either outcompeted by other types of cells or the cells rapidly downmodulate ERα expression[5]. In fact, about two-thirds of the cell line-based studies on ERα + breast cancer stem from the results of a small panel of cell lines, such as MCF7, T47D, and CAMA1[6]. Studies on the transcriptomic profiles of the clonal luminal breast cancer cell lines suggest that these cell lines do not recapitulate well the established luminal tumor subtype[7]. Therefore, widespread use of few ERα + luminal cell lines generate an information bias towards the specific clonal genetic makeup of these cell lines and their other attributes, which may not possibly apply to luminal ERα + breast cancer in general. In vivo, stable ERα expression has been reported in patient-derived xenograft (PDX) models, especially in tumor cells introduced via intraductal transplantations[8,9] and these findings have suggested a strong microenvironment-dependent dynamic component in the regulation of ERα expression.

Short-term patient-derived tumor explant culture (PDEC) systems offer potential benefits over reductionist cell cultures[10], including tumor-specific genetic and phenotypic heterogeneity and opportunities to explore tumor cell behavior within the context of authentic tumor microenvironmental components (reviewed in[10]). Patient-derived ex vivo tumor explants simultaneously provide a source of patient-specific clinical and molecular information and a live tumor sample for testing treatment options with respect to the molecular information obtained. Therefore, PDECs hold a great promise as next-generation personalized medicine tools. Unfortunately, the ex vivo tumor tissue models also commonly show a rapid loss in ERα expression in reported culture conditions[11,12]. Although there are few new ex vivo models for ERα + breast epithelial cells available[13–15], the current data provide scant mechanistic insight into the culture parameters necessary for hormone receptor expression.

Here, we report how to design and construct extracellular matrix scaffolds that conserve luminal ERα + phenotype in patient-derived human breast tissue (PDEC-N) and breast cancer (PDEC-BC) explant cultures. We show that the physiological stiffness of the culture matrix and, apparently, breast tissue microenvironment is coupled via the p38/stress-activated protein kinases-mediated stress pathway and the H3K27me3-dependent epigenetic chromatin remodeling to ERα expression in luminal breast epithelial cells and cancer cells. While the stiffest hydrogel used in this study is sufficient to maintain ERα expression in mouse-derived explants, about 20-fold higher effective stiffness is required to induce the stress and hormonal pathways in human explants. We show that ERα expression is not hardwired to luminal cell identity in breast cancer, but rather, it is an independent extracellular matrix stiffness regulated cellular pathway.

## Results

**PDEC; a patient-derived explant culture**. To establish a 3D breast cancer explant culture platform, treatment-naive fresh primary breast cancer tissue was obtained from elective breast cancer surgeries on a weekly basis. Mammary epithelial tissue from reduction mammoplasties served as the non-cancer control. One-third of each tissue sample was embedded in paraffin for immunohistochemical analyses, one-third was snap frozen for biomolecular analyses and the remainder of the sample was treated with collagenase to generate small tissue fragments (Fig. 1a, Supplementary Fig. 1a, b). These fragments were cultured in various matrices as explant cultures. The explant cultures from reduction mammoplasties were named as PDEC-N (normal) and those from breast cancer as PDEC-BC (breast cancer). In optimized culture conditions, viable cultures were established from the primary samples with a nearly 90% success rate. The present study is based on breast cancer samples from 313 patients and 123 reduction mammoplasty samples (Supplementary Data 1). According to the histopathological analysis of the pre-culture samples (example IHC shown in Fig. 1b; clinical data in Supplementary Fig. 1b, c), 86% of PDEC-BCs were luminal ERα + , which reflects the 80% incidence in newly diagnosed breast cancers in Finland and other western countries.

In Matrigel®, which is a widely used solubilized basement membrane preparation, the explants maintained their viability and structural cohesion well. On culture day 7, about 40% of the cells in both the PDEC-N and PDEC-BC explants were proliferative (Ki67-positive) (Fig. 1a; Supplementary Fig. 1g), but neither apoptotic nor hypoxic (Fig. 1a; Supplementary Fig. 1d–h). The explant sizes varied from 20 to 250 μm in diameter, with an average diameter of about 90 μm (Fig. 1a). Also, some genetic features of the original tumors were retained in the explants (Supplementary Fig. 1i).

**Matrix regulates epithelial cell identity**. In a normal breast, the epithelium throughout the ductal-lobular system is bilayered composed of an "inner" layer of cytokeratin (CK) 8/18 positive (+) luminal epithelial cells and an "outer" layer of CK14 + myoepithelial cells also known as basal cells (Fig. 1a–c). The apicobasally polarized luminal cells form contacts along their basal side with the myoepithelial cells and occasionally with the basement membrane (BM)[16]. Previous studies both by others and us demonstrated that Matrigel supports the development of normal-like basal and the luminal cell hierarchy in the cultured mouse mammary epithelial cell explants (MMECs)[17,18]. As expected, in Matrigel, the primary MMECs formed acinar structures containing a hollow lumen surrounded by inner cuboidal luminal cells and outer flat basal cells (Fig. 1c).

Surprisingly, the human PDEC-N explants failed to form such a hierarchical bilayered architecture. Upon initial culturing, PDEC-N explants retained predominantly luminal CK8 expression and contained only a few CK14 + cells (Fig. 1c). However, within 2 days of culturing, the original luminal PDEC-N started to express basal marker CK14 in the outer cells and by day 7 all cells mainly expressed basal cytokeratins (Fig. 1c). Additionally, PDEC-BCs underwent a rapid phenotypic conversion from the luminal ERα + phenotype to the basal ERα- phenotype in Matrigel (Fig. 1c; Supplementary Fig. 1j). Thus, in a standard

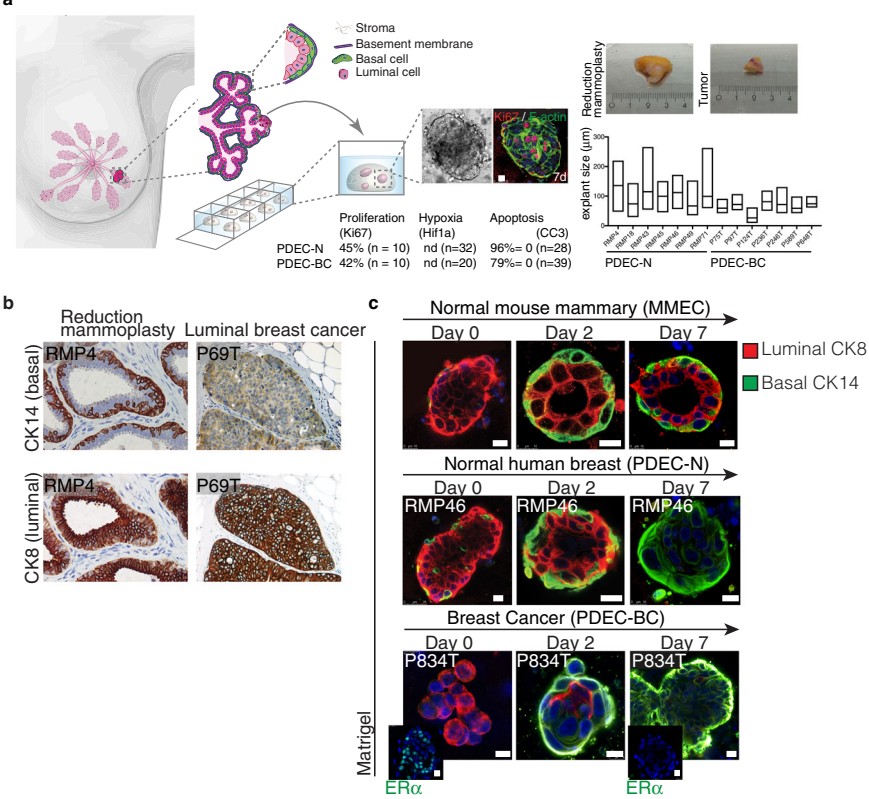

**Fig. 1 Patient-derived explant culture (PDEC) model for normal breast and breast cancer tissues. a** Schematic representation of the breast epithelial tissue (PDEC-N) and breast cancer (PDEC-BC) culture system. The table shows level of proliferation, hypoxia and viability and the bar graph shows the average explant size and the size distribution. More details of the analysis of the culture system in Supplementary Fig. 1. (nd = not detected) **b** Immunohistochemistry staining of cytokeratins 8 (CK8) and 14 (CK14) in reduction mammoplasty and luminal breast cancer samples. RMP4 refers to reduction mammoplasty corresponding to the patient number. Similarly, P69T refers to tumor sample from patient 69. N = 10 biologically independent samples. **c** Immunofluorescence images of MMECs, PDEC-N and PDEC-BC stained for CK8 and CK14 after 0, 2, and 7 days of culture in Matrigel. Day 0 samples were fixed immediately after embedding the samples in the matrix. Day 2 and day 7 indicate the culture time before fixation. Insets show immunofluorescent staining of ERα+ in the original tumor and the corresponding PDEC-BC explants, cultured in Matrigel for 7 days. N = 6 (MMEC, PDEC-N) and N = 3 (PDEC-BC) explants examined from three biologically independent samples. Scale bars = 10 μm.

Matrigel culture, PDEC-N and PDEC-BC rapidly lose the luminal epithelial phenotype and acquire the basal cell identity.

To determine the possible role of the growth matrix in the observed phenotypic conversion, we cultured MMECs and PDEC-Ns in different matrix scaffolds followed by an analysis of CK expression (Fig. 2a). Matrigel is mainly composed of BM components, whereas collagen is abundant in the stroma. These matrices contain multiple functional proteins, including latent growth factors and active adhesion molecules[19,20]. The egg white is also of an animal origin, but the heat-based polymerization of the matrix denatures most of its protein components, including the growth factors. Biopolymers such as agarose (from red seaweeds), alginate (from brown seaweeds), and a commercial animal-free matrix GrowDex®, lack cell adhesion sites and latent growth factors. These matrices were thus considered as bioinert. In one set of experiments, alginate with covalently linked RGD peptides was used to equip a biopolymeric scaffold with adhesion sites. Scanning electron microscopy (SEM) images revealed that most of the matrices consisted of fibrillar networks with a broad range in the fibril sizes and porosity (Supplementary Fig. 2a–o).

We investigated the mechanical properties of the different hydrogel matrices using oscillatory rheology, which exposed a great variation in the matrix stiffness and flow behavior under deformation, referred to as strain stiffening or strain softening (Fig. 2a; Supplementary Fig. 3a, b). Recently, we showed that agarose gels with low concentration show clear strain stiffening

behavior similar to Matrigel and collagen gels[21] (Supplementary Fig. 3a). To define the relative stiffness of each matrix, we used rheological measurements to obtain the elastic modulus (stiffness) of each gel. We note that atomic force microscopy (AFM) is one widely used technique to evaluate stiffnesses from biological surfaces, but this method is not suited to estimate the elastic properties of larger gel volumes and the stiffness values obtained via AFM or other similar techniques cannot be directly cross-referenced with the metrics obtained via rheological measurements[22]. Therefore, the metrics provided in this study should be considered only as a technical parameter that allows side-by-side comparisons of different matrix stiffnesses and our stiffness values cannot be compared with the stiffness values obtained via AFM or other similar techniques.

In Matrigel both the MMEC and PDEC-N acquired the full basal identity (CK14 + ), whereas in collagen and GrowDex, only a partial phenotypic switch occurred, as indicated by the presence of CK14 + ; CK8 + cells (Fig. 2a). In contrast, no phenotypic switch was observed in alginate, agarose, or egg white (CK8 + ). Thus, we termed alginate, agarose, and egg white as the luminal identity-preserving matrices (LMx) and Matrigel as a basal identity-promoting matrix (BMx). In the quantitative analyses, the luminal-to-basal phenotypic switch occurred in all PDEC-N (6 out of 6) and most (6 out of 8) of the PDEC-BC cultures in BMx, whereas all cultures retained the luminal identity in LMx (Fig. 2b, c; Supplementary Fig. 3c). Together, these experiments identified

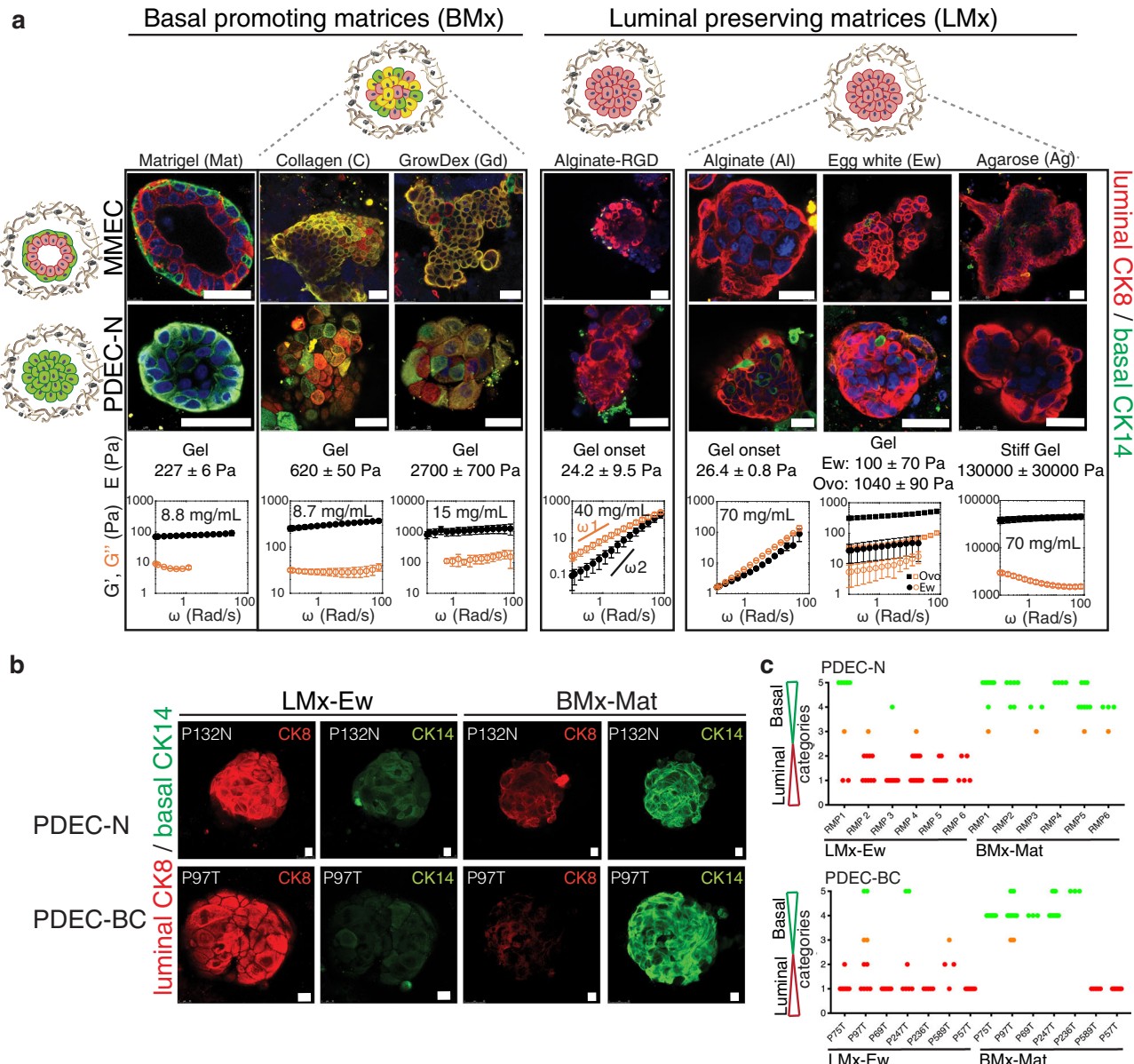

**Fig. 2 Matrix defines the epithelial cell identity. a** Immunofluorescence images of MMECs and PDEC-N stained for CK8 and CK14 after 7 days culture in the indicated matrices. The basal cell identity promoting matrices were named as BMx. These matrices included Matrigel (Mat), collagen (C), and GrowDex (Gd). The luminal cell identity preserving matrices were named as LMx. They included alginate (+/- RGD peptide), agarose (Ag), and egg white (Ew). Ovo denotes ovomucin, which is the gelling component of egg white. The phenotype of ovomucin grown explant was identical to egg white grown explants. The mechanical properties of the matrices were measured with oscillatory rheology: The storage modulus ($G'$, black filled symbol) at different oscillation frequencies ($\omega$) describes the elastic, solid-like properties, and the loss modulus ($G''$, orange open symbol) illustrates the viscous, liquid-like, properties of the material. $G'$ and $G''$ together express whether the matrix is an elastic gel or a viscous fluid when measured as a function of the oscillation frequency. In general, the material is gel, if $G' > G''$, and both $G'$ and $G''$ are nearly independent of the frequency. If material is viscous fluid $G'' > G'$ and it has strong scaling with frequency (at low frequencies $G' \sim \omega^2$ and $G' \sim \omega^1$). The 3D matrices used in this study varied from viscous fluid ($G' \sim 1\,\text{Pa}$) and soft gels ($G' \sim 10\,\text{Pa}$) all to way to stiff gels ($G' \sim 10{,}000\,\text{Pa}$). The rheological frequency sweeps show that where egg white, and agarose are gels, the alginates are viscous fluids in PBS. Elastic modulus ($E$) is estimated from complex modulus ($G^*$), $E = 2(1+\upsilon)G^*$, with an assumed Poisson's ratio of $\upsilon = 0.44$. For comparison to the day 0 sample, see Fig. 1c. **b** Immunofluorescence images of PDEC-N and PDEC-BC stained for CK8 and CK14 after culture in LMx-Ew or BMx-Mat (7d). **c**, Quantification of the luminal and basal cytokeratins in PDEC-N and PDEC-BC explants. For details regarding the quantification, see Supplementary Fig. 3c. $N = 6$ (PDEC-N) and $N = 7$ (PDEC-BC) independent experiments. All data are presented as mean values +/- SD. Scale bar = 10 μm.

several matrix scaffolds capable of supporting the luminal identity in normal breast and breast cancer explant cultures.

**Matrix alters transcriptomic profiles in the explant cultures.** Total RNA sequencing was performed to determine the gene

expression patterns in MMECs, PDEC-N, and PDEC-BC explants grown in LMx or BMx matrices (Fig. 3a). First, MMECs were cultured for 1 week in BMx-Mat (Matrigel) or in LMx-Al (alginate). Principal component analysis (PCA) revealed a tight clustering among independent samples according to the growth matrix, indicating a strong matrix-dependent component in the

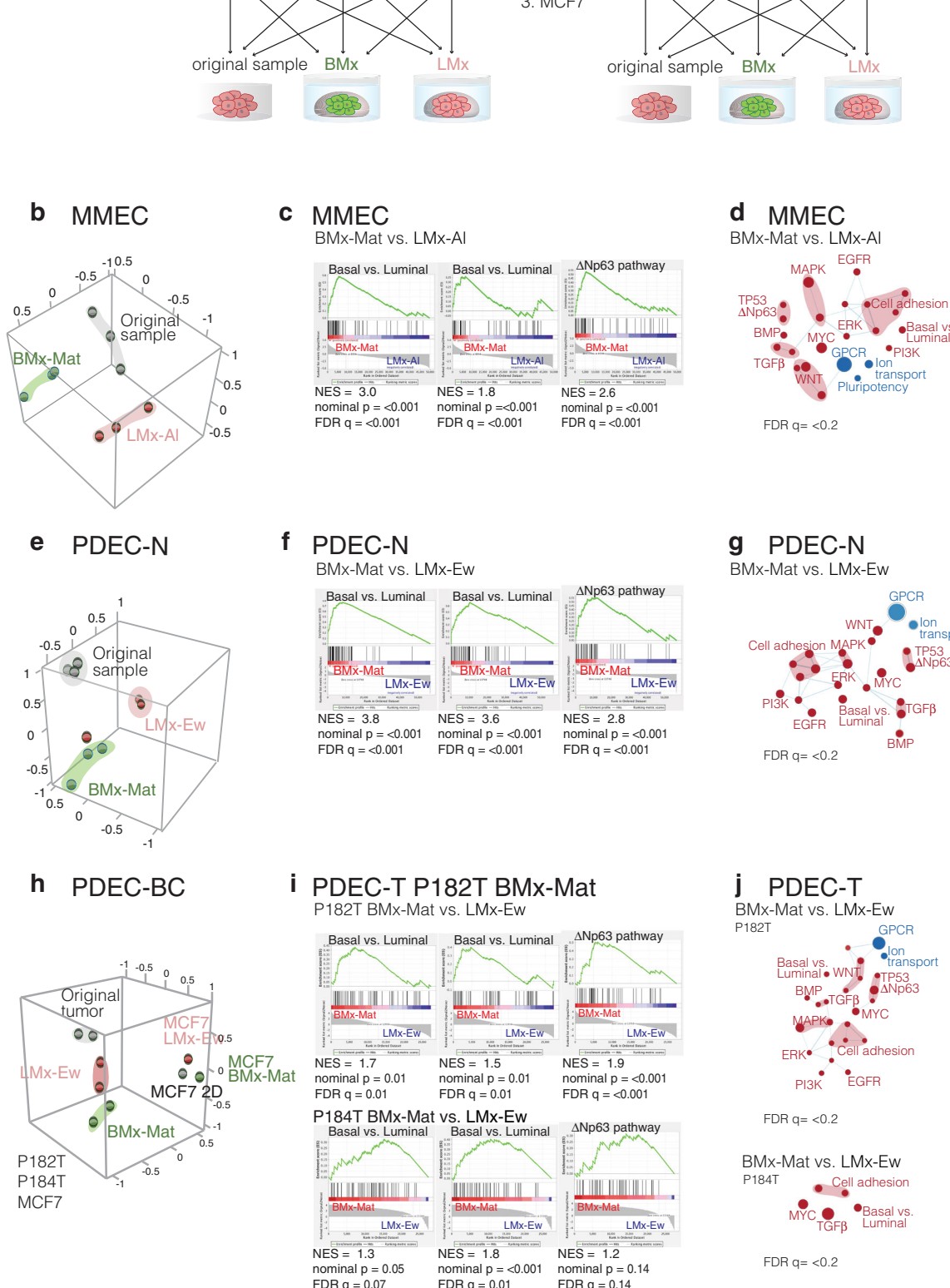

gene expression profile of MMECs (Fig. 3b). The Gene Set Enrichment Analysis (GSEA) indicated that the enriched gene expression patterns in BMx-grown MMECs were faithful to the basal identity-associated genes (Fig. 3c, d)[23–27]. Total RNA sequencing analysis was also performed on three PDEC-N samples yielding results similar to MMECs; the human samples

clustered according to the BMx and LMx gels and the basal identity-associated gene expression signatures were enriched in BMx-grown PDECs (Fig. 3e–g). We analyzed the impact of the matrix on the transcriptomes of breast cancer-derived PDEC-BC cultures and included in the analysis two luminal ERα + tumors (P182T, P184T) and one estrogen receptor-positive breast cancer

**Fig. 3 Matrix alters transcriptomic profiles in the explant cultures. a** Experimental design. **b**, Principal components analysis (PCA) showing the matrix-dependent clustering of the MMECs. Mammary epithelial tissue samples were collected from three different mice and each sample was divided into three parts. One part remained uncultured (grey), the second part was cultured in BMx-Mat (green), and the third part was cultured in LMx-Al (pink) for 7 days. **c** The gene-set enrichment analysis (GSEA) shows the enrichment of the basal epithelial cell identity-associated gene sets in the explants grown in BMx matrix. **d** Cytoscape's Enrichment map shows enrichment of basal phenotype-associated gene-sets in BMx-cultured explants as compared with LMx-cultured explants. Node size: number of genes in the signature; node color: red—enrichment in BMx-Mat vs LMx, blue—underrepresented in BMx-Mat. See the full maps in Supplementary Fig. 4. **e, f**, PCA, GSEA, and enrichment map similar as in **b–d** for PDEC-N **e–g**, and for two PDEC-BCs (P182T, P184T) and MCF7 breast cancer cells grown in 2D **h–j**. Abbreviations: NES: normalized enrichment score, FDR q: false discovery rate.

cell line (MCF7). In PCA, both tumor samples clustered separately (Fig. 3h). As with PDEC-N, the gene expression signatures of the breast cancer explants remained faithful to the cell identity (Fig. 3i, j).

**Matrix stiffness regulates ERα expression.** Historically, it has been exceedingly difficult to retain ERα expression in breast cancer cell cultures. Even in short-term cultures of freshly isolated fragments of breast tissue such as PDEC-N, the hormone receptor expression is lost[11,12]. We tested two LMx matrices—LMx-Al and LMx-Ew—for their capacity to sustain ERα expression. While both matrices preserved the luminal cell identity ex vivo, neither of these matrices could sustain the ERα expression (Fig. 4a, in Fig. 4b note the negative NES values). Therefore, the luminal identity features and ERα are independently regulated in a 3D culture.

Among the three LMx matrices identified in this study, LMx-Ag was over three orders of magnitude stiffer than the relatively soft LMx-Al and LMx-Ew matrices (Figs. 2a, 4c; Supplementary Fig. 3a, b). Curiously, when the global genome expression profiles were compared between LMx-Ag and LMx-Al cultured MMECs, only the former expressed a high ratio of exonic-to-intergenic sequences along with the luminal identity (Fig. 4c). The difference was striking, since explants from the same original tissue piece expressed up to 68% of the exonic reads in LMx-Ag as compared to only 4% in the LMx-Al matrix. The high exonic-to-intergenic sequence ratio in the LMx-Ag-grown MMECs was similar as in the original uncultured sample (50% exonic vs. 19% intergenic). In both PCA and the gene expression profiling, LMx-Ag-grown MMECs clearly clustered separately from LMx-Al (Fig. 4c, d; Supplementary Fig. 3d) with little overlap in the gene expression profiles (Fig. 4e).

A closer inspection of the LMx-Ag-enriched pathways revealed the presence of estrogen response mRNA signatures as well as profiles indicating ERα binding (genomic ERα, intracellular steroid hormone receptor signaling pathway, estrogen receptor binding; Fig. 4e–g). Subsequent inspection of the ERα protein using immunofluorescent staining methods exposed a nuclear localization of ERα specifically in the explants grown in the stiff LMx-Ag matrix (Fig. 4g). Thus, we identified one growth matrix that preserved luminal identity and ERα expression and the data suggest that sufficient stiffness may be required for nuclear ERα expression.

To further explore the significance of the matrix stiffness to ERα expression, we prepared LMx-Ag gels in a gradually increasing polymer concentration (1–7% w/v). The matrix stiffness increased proportionally as evidenced by the rheological measurements (Fig. 4h; Supplementary Fig. 3b). While ERα was absent in the soft matrices (1 and 2%, $G' < 10$ kPa), the nuclear ERα was clearly expressed in the stiff LMx-Ag matrices (3–7%, $G' > 10$ kPa; Fig. 4i). Thus, matrix stiffness appears to represent a critical requirement for ERα expression with a storage modulus ($G'$) threshold of 10 kPa for MMECs, corresponding to the elastic modulus (E) of 20 to 30 kPa depending on the Poisson's ratio

(between 0 to 0.5), which is ideally 0.5 for incompressible materials such as rubber.

For the functional validation of transcriptionally active ERα in the mouse explants, we examined the effect of the antiestrogen treatment on the ERα target gene sets and individual target genes (Fig. 4k–l; Supplementary Fig. 3e). We treated 7% LMx-Ag cultured MMECs with standard drugs for anti-estrogen treatment, tamoxifen and fulvestrant, along with three other selective estrogen receptor modulators (SERM) or degraders (SERD) (Fig. 4j–l; Supplementary Fig. 3e, f). In the GSEA analysis comparing the control and treated explants, the ERα-regulated gene sets were clearly diminished in the treated samples (Fig. 4k). Furthermore, we validated the effect of different antiestrogen treatments on the mRNA expression level of the ERα downstream targets the progesterone receptor (PGR) and GREB1, finding that the expression was consistently downregulated in the treated explants (Fig. 4l; Supplementary Fig. 3e). These results demonstrated the presence of functional ERα in stiff (7%) LMx-Ag-cultured mouse explants.

**Stress signaling is required for ERα expression.** While the LMx-Ag matrix retained the nuclear ERα in MMECs, it failed to do so in the human PDEC-N and PDEC-BC cultures (Fig. 5a, b). Moreover, the LMx-Ag matrix failed to enrich the exonic reads as it did in MMECs (compare Figs. 5c and 4c). To understand why human nuclear ERα was not retained in LMx-Ag, we explored the pathway signatures that were clearly different between LMx-Ag-cultured mouse and human explants. In parallel, we tested a set of growth factors and pathway-targeted drugs for their ability to maintain ERα in PDEC cultures (Table 1). Interestingly, in MMECs, the stress pathway signature was enriched in the uncultured tissue samples and in the LMx-Ag-grown explants when compared to the explants grown in LMx-Al (Fig. 4e; Fig. 5d). The stress pathway was also enriched in the LMx-Ag grown MMECs when compared to LMx-Al grown MMECs (Supplementary Fig. 3g). In contrast to MMECs, the stress pathway was enriched in uncultured human PDEC-N and PDEC-BC explants when compared to the LMx-Ag-grown explants (Fig. 5d). Together, these results suggest that the MMECs experience similar level of stress in LMx-Ag matrix as in the uncultured state. However, for human PDEC-N and PDEC-BC explants the LMx-Ag matrix falls short in imposing the same level of stress that is present in the uncultured tissue (Fig. 5d; see Supplementary Fig. 5c for extended analysis of PDEC-BC). To explore if a chemically induced stress pathway induces ERα in the explant cultures, we administered anisomycin, which is a potent activator of stress-activated MAP kinases (SAPKs) and p38 MAP kinase to the explant cultures. Strikingly, anisomycin induced the strong expression of ERα in MMECs, PDEC-N, PDEC-BC, and TNBC cell lines (Fig. 5e, f; Supplementary Fig. 5d, e).

**EZH2-dependent histone 3 trimethylation represses ERα expression.** In the MMECs, high relative expression ratio of exon sequences to intergenic region (ig) sequences associated with ERα

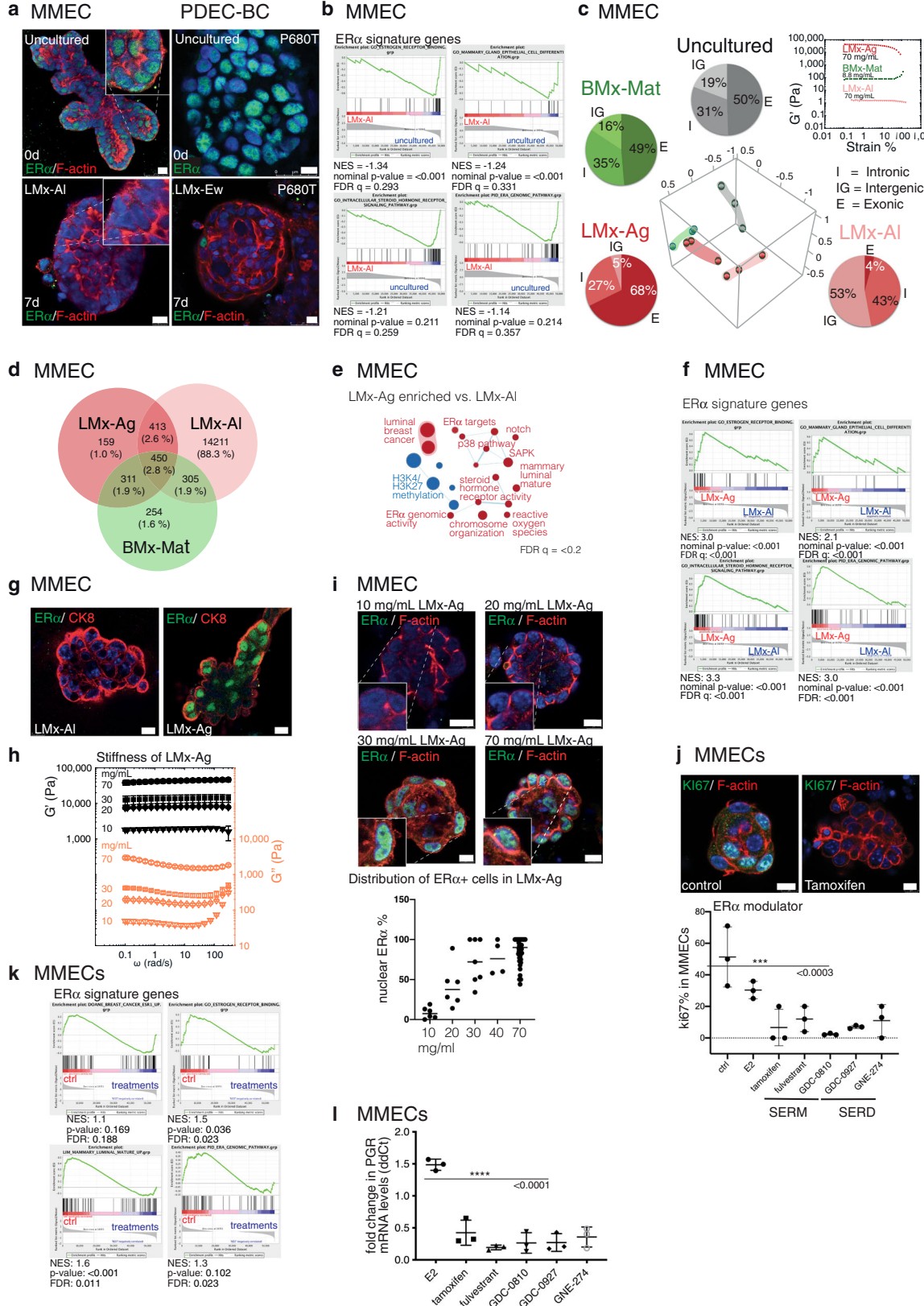

expression in LMx-Ag cultured explants whereas low proportion of exon sequences associated with ERα- phenotype in LMx-Al cultures; in human cultures all matrices failed to support high proportion of exon expression and ERα expression (compare Figs. 4c and 5c). Interestingly, earlier studies have suggested that the switch from high intergenic/intronic sequence expression pattern to exon-sequence dominated expression pattern is coupled to epigenetic reprogramming during the stem cell differentiation[28]. Since the pluripotency related signatures were also specifically found enriched in both mouse and human explant cultures grown in the soft, non-stressing, ig-expression enriching and non-ERα supporting matrices (Supplementary

**Fig. 4 Matrix stiffness regulates ERα expression in MMECs. a** Immunofluorescence images of MMECs and PDEC-BCs stained for ERα and filamentous actin after 0 or 7 days in an LMx-matrix. **b** GSEA analysis of ERα signaling signature in LMx-Al cultured MMECs (7d) compared to uncultured samples. **c** PCA of RNAseq data obtained from MMECs cultured 7 days in indicated matrices. Pie charts show the relative distribution of exonic (E), intronic (I) and intergenic (IGR) transcripts in the RNA sequencing. Rheological strain amplitude-sweep measurements show stiffness of the examined matrices. **d** The Venn diagram illustrates the number of transcripts specific for the explants grown in indicated matrices. **e-f** Enrichment of the ERα activity-related gene expression signatures in the stiff LMx-Ag matrix. **g** Immunofluorescence images of ERα and CK8 expression in MMECs grown in soft LMx-Al and stiff LMx-Ag matrix. **h** Rheological frequency sweeps of LMx-Ag show the increasing stiffness (storage modulus, $G'$ shown by black-filled symbols and the loss modulus $G''$ shown by orange-open symbols) according to the increasing polymer concentration (10, 20, 30, and 70 mg/mL). $N = 3$ independent experiments, except $N = 2$ with 20 mg/mL. **i**, Immunofluorescence images show MMECs, grown in indicated concentrations of LMx-Ag (7d) and stained for ERα and F-actin. The graphs show quantification of the fraction of ERα-positive cells compared to total cell number ($n = 66$ explants from six different mice). **j** The effect of tamoxifen on proliferation (Ki67) in LMx-Ag grown explants. The graph shows percentage of the proliferating cells compared to the total cell number of explants after SERM/SERD treatment. Statistical significance was tested with one-way ANOVA with Dunnett's multiple comparisons post hoc test. Significance for multiple comparisons: ***$p = 0.0004$ tamoxifen, **$p = 0.0013$ fulvestrant, ***$p = 0.0002$ GDC-0810, ***$p = 0.0005$ GDC-0927, **$p = 0.0011$ GNE-274. **k** GSEA analysis show a suppression in the ERα activity-related gene expression signatures after treatment with SERM/SERD compounds. **l** QRT-PCR for progesterone receptor (*PGR*) after SERM/SERD treatment in MMECs. Statistical significance was tested with a one-way ANOVA with Dunnett's multiple comparisons post hoc test: ****$p < 0.0001$. All data are presented as mean values +/- SD. Scale bar = 10 μm.

Fig. 3h–k), we examined whether epigenetic silencing could explain the loss of ERα expression in explant cultures. Interestingly, we found that the gene expression signatures corresponding to gene-repressive H3K27me3 histone methylation pathway were specifically enriched in those MMEC culture conditions, which failed to sustain ERα and in all human explant cultures (Fig. 6a; Supplementary Fig. 5a–c). Intriguingly, our survey of the published data on epigenetic modifications at *Esr1* promoter (Cistrome database[29]) revealed prominent H3K27me3 peaks in the *Esr1* promoters of four TNBC cell lines (MDA-MB-231, MDA-MB-436, MDA-MB-453, SUM159PT), while similar peaks were not observed in the five analyzed ERα + cell lines (MCF7, T47D, UACC812, ZR-75-1, ZR-75-30) (Supplementary Figure 8e)—further pointing to epigenetic mechanisms in downregulation of ERα[30–35].

In mammalian cells, the primary catalyst of H3K27me3 trimethylation is the enhancer of zeste homolog 2 (EZH2), which acts as a catalytic subunit of the polycomb repressive complex 2 (PRC2)[36]. We used GSK-126, a highly selective EZH2 inhibitor[37], to explore the possible functional involvement of the H3K27me3 pathway in ERα regulation. We found that the inhibition of EZH2 efficiently restored nuclear ERα expression in LMx-Al-cultured MMECs, TNBC cell line aggregates, and both PDEC-N and PDEC-BC explants (Fig. 6b, c). Moreover, as a functional validation of a transcriptionally active ERα in the stressed or H3K27me3 pathway-inhibited explants, we demonstrate that both anisomycin and GSK-126 treatment upregulate ERα downstream targets, the progesterone receptor (PGR) and GREB1 in PDEC-BC (Fig. 6c). The stress mediator p38 has been previously shown to phosphorylate EZH2 and inhibit its activity[38]. In agreement with earlier notions associating stress with inhibition of EZH2, anisomycin not only induced the stress pathway and upregulated ERα, but also diminished the H3K27me3 histone marks (Fig. 6d). Together, these results suggest that the physiological stiffness of the tissue microenvironment regulates ERα expression via the stress pathway and H3K27me3-dependent epigenetic chromatin remodeling (Fig. 6e).

**Compression induces ERα expression in human explants.** While a stiff matrix was sufficient to maintain ERα expression in the MMECs, the stiff LMx-Ag–grown human explants failed to retain ERα expression without the implementation of chemical stress or inhibition of EZH2 in the cultures. Therefore, a stiff matrix is insufficient alone to upregulate human ERα expression or, alternatively, human explants might require a higher pressure than mouse explants to activate stress signaling and ERα expression. In support of the latter hypothesis, the mainly fat-containing mouse mammary fat pad is biologically less stiff microenvironment for epithelial glands than the fibroblast-enriched human breast stroma (Supplementary Fig. 9)[39]. Moreover, breast cancer cells, which often show the widespread expression of ERα (grade 3: 75–100%) generally reside in a stiffer environment (the tumor) than breast epithelial cells of the normal gland (Supplementary Fig. 9). Therefore, it is possible that breast cancer cells experienced less pressure in our stiffest matrix (7% LMx-Ag) than in the authentic tumor tissue (Supplementary Fig. 9).

To impose a higher pressure for human breast cancer explants than that attained with the matrix only, we exposed the LMx-Ag-cultured PDEC-BC and PDEC-N to an enhanced physical compression, generated by the magnetic force. A metal grid was placed on top of the cultures and two magnets were placed on the opposite sides of the cultures to generate the compression (Fig. 6f). The magnets compressed the cultures by 37 kPa, which resulted in a 178 % increase to the initial average shear modulus, $|G*|$, of 7% LMx-Ag. Particularly, the effective $|G*|$ increased from the initial value of 47 kPa (uncompressed) to 129 kPa (compressed). The uncompressed and compressed conditions corresponded to $E = 134$ kPa and $E = 373$ kPa in terms of elastic modulus. In contrast to the uncompressed conditions, under the compression, ERα was expressed, p38 was phosphorylated and the ERα pathway genes were responsive to tamoxifen, indicating appearance of stress and a functional ERα pathway (Fig. 6g, h; Supplementary Fig. 5f). These results demonstrate that matrix stiffness regulates stress signaling and ERα expression also in the human breast tissue- and breast cancer-derived explants.

To test the functional importance of p38 mediated stress pathway for ERα expression, we chemically inhibited p38 in LMx-Ag cultured MMECs and magnet compressed PDEC-BCs (for validation of p38 MAPK inhibitors, see Supplementary Fig. 6a). As evidenced by the western blot, RNA sequencing, and immunofluorescence microscopy analysis, the inhibition of p38 abolished the nuclear ERα expression (Fig. 7a–d) and suppressed the ERα activity in both MMEC and PDEC-BC (Supplementary Fig. 6b, d).

In addition, consistent with our earlier notion suggesting involvement of p38 as a mediator of the matrix stiffness to EZH2-mediated trimethylation of H3K27 and downmodulation of ERα activity, inhibition of p38 activated EZH2 (negative phosphorylation of EZH2-p (T367) diminished) and resulted in enhanced trimethylation of H3K27 (Fig. 7b, d). When we applied p38 and EZH2 inhibitor together, H3K27me3 did not increase and ERα expression was not downmodulated (Fig. 7e, Supplementary Fig. 6e).

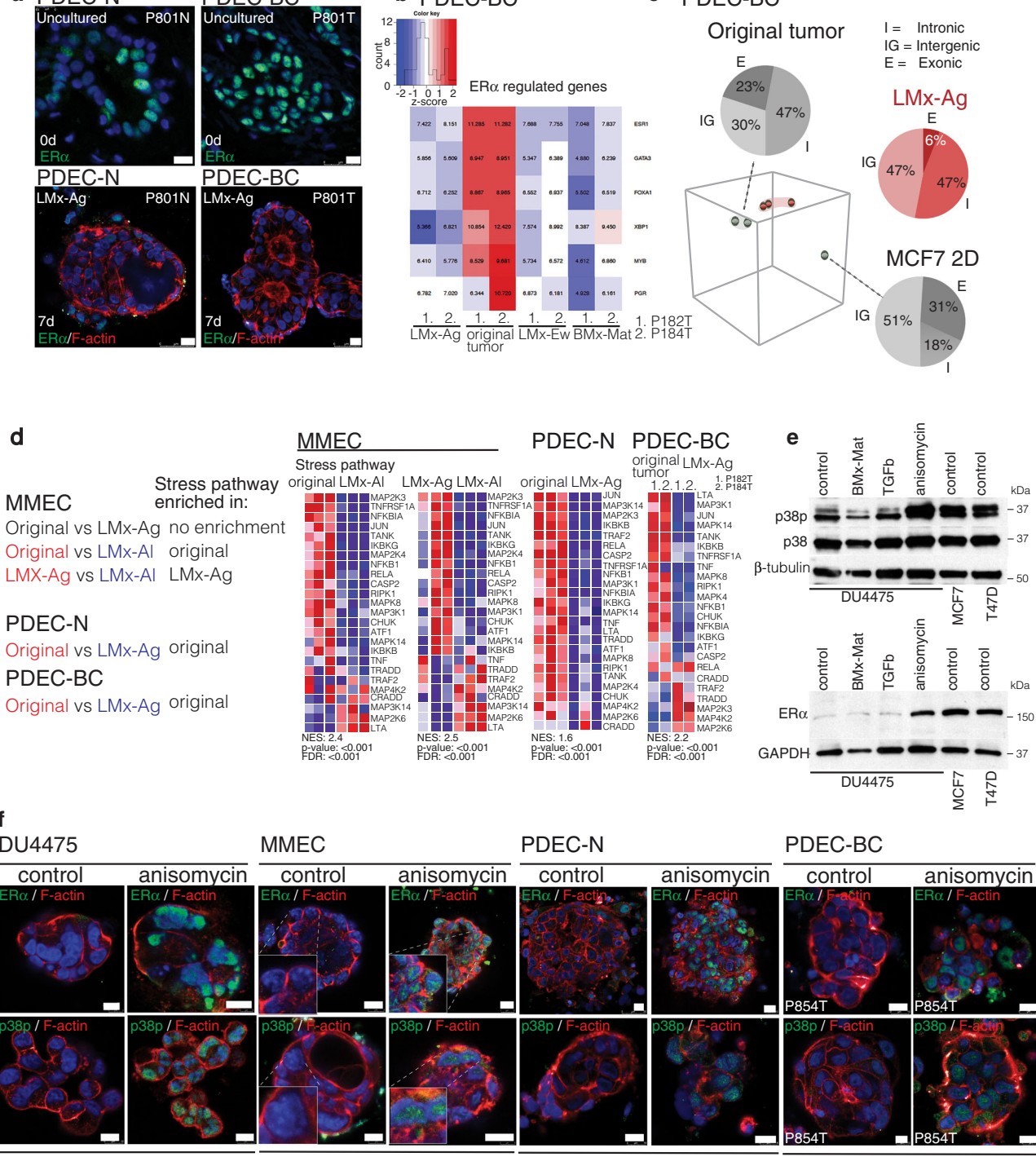

**Fig. 5 Stress signaling is required for ERα expression. a** Immunofluorescence staining of ERα (green) in uncultured PDEC-N and PDEC-BC samples and after 7 days in a LMx-Ag matrix. $N = 5$ explants examined from 3 biologically independent samples. **b** The heatmap shows the expression of ERα-regulated gene sets in PDEC-BCs (P182T and P184T) in different matrices in comparison to the uncultured original samples. **c** PCA of RNAseq data obtained from PDEC-BC and MCF7 cells cultured in LMx-Ag matrix (red) for 7 days compared to the uncultured original tumor / 2D cultured MCF7 cells (grey). Pie charts show the relative distribution of exonic (E), intronic (I) and intergenic (IGR) transcripts in the RNA sequencing. **d** Heatmaps from GSEA analysis show the enrichment of stress pathway in MMECs, PDEC-N and PDEC-BCs. Different comparisons and the corresponding enrichments are shown in left. **e** Western blot analysis shows the effect from anisomycin treatment on p38p/p38 and ERα expression in the DU4475 TNBC cell line. TGFβ (2 ng/ml) serves as the negative control, while MCF7 and T47D are positive controls for the ERα ($n = 3$). **f**, Immunofluorescence staining of p38p in DU4475 cells, MMECs, PDEC-Ns, and PDEC-BCs in control and after anisomycin treatment. $N = 6$ explants examined from three biologically independent samples. Scale bar = 10 μm.

**Table 1 The table represents a list of compounds tested in ERα activation.**

| Compound | Provider | Mode of Action | Concentration |
|---|---|---|---|
| 17β-estradiol | Sigma-Aldrich/Merck | | 0.1–10 nM |
| IGF-1 | Sigma-Aldrich/Merck | | 5 ng/ml |
| Ryanodine | Tocris Bioscience | Ca2+ release inhibitor | 100–100 mM |
| BAPTA-Am | Abcam (ab120503) | Cell permeant Ca2+ chelator | 1–10 mM |
| MK-2206 | ChemieTek | AKT inhibitor | 10 nM |
| MK-0752 | Selleckchem | Gamma secretase /notch inhibitor | 5–10 nM |
| Pictilisib | LC Laboratories | PI3K inhibitor, pan-class I B kinase inhibitor | 10 nM |
| Prolactin | Biotechne | | 100 ng/ml |
| GSK-343 | Sigma-Aldrich/Merck | EZH2 inhibitor | 100–3 mM |
| Estriol | Sigma-Aldrich/Merck | | 100 mM |
| Dantrolene | Santa Cruz | Ca2+ release inhibitor | 50 mM |
| SP600125 | Sellechem | inhibitor | 10 μM |
| RWJ67657 | Sellechem | p38 inhibitor | 10 μM |
| SB203580 | Sellechem | p38 inhibitor | 20 μM |

Together, since phosphorylation of EZH2 at T367 suppresses the protein activity, our results from both mouse and human explant cultures altogether are consistent with a mechanistic model presented in Fig. 6e. Accordingly, specifically a stiff matrix induces p38 mediated stress pathway, which keeps EZH2 phosphorylated at T367 thus suppressing the activity of this key enzyme that catalyzes the addition of methyl groups to histone H3 at lysine 27. In the absence of EZH2 activity (H3K27me3 low), the expression of ERα is favored whereas in the presence of EZH2 activity (H3K27me3 high), the expression of ERα is downmodulated.

We also explored the involvement of JNK, which is another key stress-activated protein kinase (SAPK), using a specific JNK inhibitor (SP600125). However, our experiments did not find a role for JNK in mediating the stiff matrix dependent expression of ERα or influencing p38/EZH2/H3K27me3 activity in LMx-Ag cultured MMECs or magnet compressed PDEC-BCs (Fig. 7a–d). Also, no alteration in the ERα regulated gene sets were observed after JNK inhibition although JNK regulated gene sets were clearly suppressed (Supplementary Fig. 6c).

**Matrix stiffness coupled ERα expression in breast cancer and tissue.** To determine whether the p38 mediated stress signaling might associate with the ERα status in clinical samples of breast cancer, we analyzed 42 invasive breast cancer samples for phospho-p38 and ERα expression by immunohistochemistry (Fig. 7f; Supplementary Figure 6f & 7a, b). The association between phospho-38p and ERα expression was statistically significant (Pearson's product-moment correlation 0.98). These data are in line with several earlier studies suggesting significantly higher expression of p38p in the ERα + breast tumors[40–42].

Additionally, we investigated whether the expression of p38 might associate with ERα in the breast cancer patient samples in the Cancer Genome Atlas (TCGA; invasive breast carcinoma dataset for 892 RPPA samples). We observed a positive correlation in the sample-to-sample data between p38 and ERα protein expression levels (Supplementary Fig. 8a, for further analysis of associations between ERα, p38 and p38 upstream kinases MAP3K1 and MAP2K3 see Supplementary Fig. 8b–d). Notably, consistent with our hypothesis that EZH2 acts as a negative regulator of ERα, we observed a negative correlation between the mRNAs of ERα/PGR and EZH2 in the METABRIC dataset (Supplementary Fig. 8c). The negative correlation between EZH2 and ERα, is also consistent with the earlier studies, that describe increased levels of EZH2 in ERα negative breast cancer[38,43,44]. Altogether, the data support a role for phospho-

p38, its upstream MAPK pathway, and EZH2 in the regulation of ERα expression in normal breast and breast cancer.

Finally, we sought to find evidence to support the role of matrix stiffness in the ERα expression in intact human breast. For this purpose, we investigated the possible association between the mammographic breast density (MBD) and ERα expression. The high MBD reflects a greater amount of glandular and connective tissue compared to the fat as well as enhanced tissue stiffness[45–47]. Furthermore, women with the highest MBD exhibit a four- to sixfold increase in breast cancer risk compared to women with nondense breasts[48–50]. Preoperative mammography is performed prior to noncosmetic reduction mammoplasty in Finland, and, therefore, each reduction mammoplasty (RMP) sample in our series could be annotated with a pre-existing clinical MBD score (for the clinical data, IHC stainings, and scoring, see Fig. 7g; Supplementary Fig. 6g, h). We immunostained histological sections of 18 RMP samples for ERα expression and plotted the ERα expression on a scale from 0 to 4 against the breast density scores defined via the Breast Imaging Reporting and Data system (BI-RADS) (Supplementary Fig. 6g). The results demonstrate a significant correlation between the ERα expression score and the mammographic density, supporting the notion that ERα expression is regulated via mechanosensing pathways in the breast (Fig. 7g). The current results from ex vivo culture studies are summarized in Fig. 7h and Supplementary Fig. 9.

**Discussion**

The current study presents 3D tissue culture conditions, which conserve the luminal ERα + epithelial phenotype in patient-derived breast tissue and breast cancer explants. We show that the epithelial cell identity is not a stable feature in a culture but highly sensitive to changes mediated by the matrix environment. Only by varying the matrix component we could generate an entire range of different mammary cell identities from the basal phenotype to the luminal ERα- and luminal ERα + phenotypes. In Matrigel, mammary epithelial tissue explants underwent a rapid phenotypic switch from the luminal to the basal cell identity. However, we observed species-specific differences; MMECs formed normal-like bilayered epithelial structures with the basal cells facing the matrix and the luminal cells forming the inner layer. In contrast, the non-cancerous human mammary epithelial cells (PDEC-N) primarily assumed the basal phenotype (schematic representation of the matrix effects in Fig. 7f). A similar phenotypic switch occurred in most, but not in all breast cancer explants (PDEC-BC), perhaps implicating confounding genetic

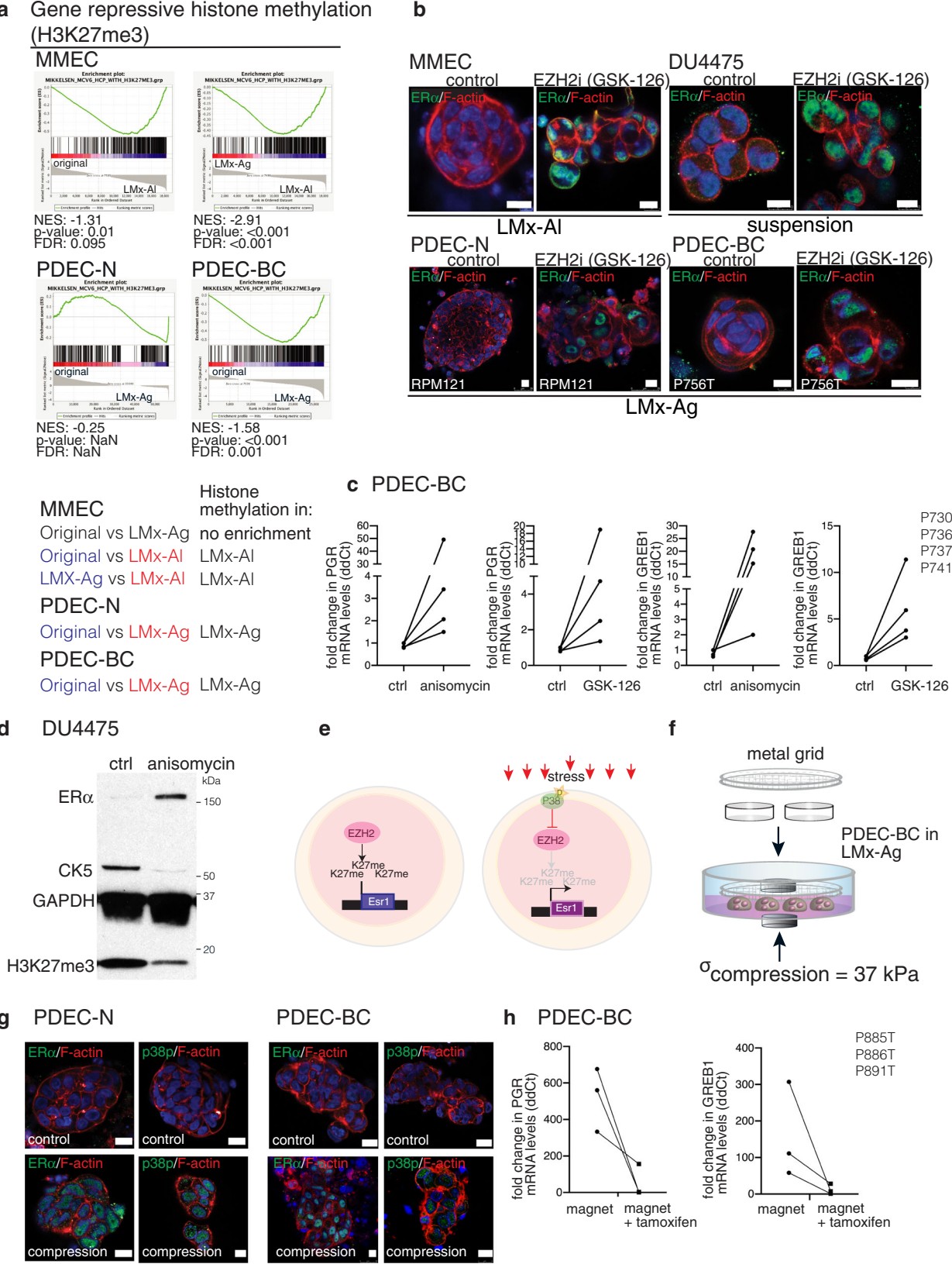

alterations that interfere with matrix-dependent plasticity. Interestingly, earlier studies have demonstrated a microenvironment-dependent conversion of the basal-like breast cancer phenotype to ERα + phenotype in vivo via carcinoma-associated fibroblast-dependent mechanism, although the role of matrix stiffness in this context still remains to be clarified[51].

Importantly, the originally luminal breast tissue or breast cancer samples retained the luminal phenotype in three bioinert LMx scaffolds. However, even if the cells retained luminal identity, different matrices had a strikingly different impact on the global regulation of genome expression. While the soft alginate matrix (LMx-Al) supported the luminal phenotype with a

**Fig. 6 Compressive stress and EZH2-dependent histone 3 trimethylation regulate ERα expression. a** Enrichment of the gene-repressive histone trimethylation (H3K27me3) signatures in different MMECs, PDEC-N, and PDEC-BC samples. Different comparisons and the resulting enrichments are shown below of the graphs. **b** Immunofluorescent staining of ERα expression in control and enhancer of zeste homolog 2 (EZH2) inhibitor GSK-126 treated explants from DU4475 cells, MMECs, PDEC-N, and PDEC-BC. N= 6 explants examined from three biologically independent samples. **c** PDEC-BCs from four patients (PxxxT) were exposed to anisomycin or GSK-126 and the effect on ERα-regulated *GREB1* and *PGR* genes was measured using QRT-PCR. **d** Western blot analysis shows ERα, CK5 and H3K27me3 expression in the DU4475 cells after 48 h treatment with anisomycin (*n*= 3). **e** A model for the stress-mediated regulation of ERα expression. **f** Illustration of the magnetic cylinder-mediated compression method. **g** Immunofluorescence images of PDEC-Ns and PDEC-BCs in the LMx-Ag following overnight compression with the magnetic cylinders, stained as indicated. N = 4 explants examined from 3 biologically independent samples. **h** PDEC-BCs from three patients were exposed to magnet mediated compression for 48 h and with or without tamoxifen treatment. The effect on ERα regulated *GREB1* and *PGR* genes was measured with QRT-PCR. Scale bar = 10 μm.

strikingly low (4%) contribution of sequence reads from the exonic sequences, the stiff LMx-Ag matrix supported >50% exonic sequence representation, which figure was similar to the uncultured tissue sample. Altogether, the soft gel promoted the intergenic and intron enriched global gene expression pattern that is characteristic for stem cells[28], it enriched stem cell and pluripotency-related gene expression signatures and induced gene repression related H3K27me3 methylation profiles. In association with these genetic changes, the soft gel failed to induce p38 stress pathway and ERα expression in MMECs. The stiff matrix (LMx-Ag) had opposite impact on MMECs; it induced the p38 stress pathway, the ERα expression and >50% exonic sequence representation but it did not induce stem cell/pluripotency-like gene expression patterns or H3K27me3 associated gene repressive activity. These results would be consistent with a model that the extracellular matrix-dependent compressive forces are constantly signaling to the cells via the cellular p38-mediated cellular stress pathway and this activation of p38 supports the exon-enriched gene expression pattern typical for differentiated cells, also including ERα expression. According to this model, a soft microenvironment signals to opposite direction, favoring loss of the differentiated phenotype and downmodulation of ERα. Interestingly, during the mammary gland development, the pluripotent ERα- CK8 + cells have a capacity to differentiate into multiple different lineages, including differentiated ERα + CK8 + cells[52] and there are multiple factors affecting to the stiffness in vivo, such as extracellular matrix composition, fibroblasts or myoepithelial cells. It remains a challenge for future studies to define whether the developmentally regulated processes involve compressive forces to regulate ERα expression.

Interestingly, a chemically induced stress by anisomycin phenocopied the effect of stiff microenvironment in the explant cultures. Like stiff gels, anisomycin treatment induced p38 stress pathway, repressed H3K27 trimethylation and upregulated ERα. While our current studies did not explore the exact biochemical mechanisms how p38 mediated phosphorylation of EZH2 impacts H3K27 trimethylation at *Esr1* locus, we wish to point out earlier studies, which have suggested that p38 can directly regulate EZH2 through phosphorylation, preventing EZH2 from entering the nucleus and, thus, inhibiting EZH2's nuclear methyltransferase activity[38,43]. Such linear p38-EZH2 pathway could explain the ERα regulation observed in the current study (Fig. 6e). In support, we show that the inhibition of p38 simultaneously prevented the phosphorylation of EZH2 and led to an increase in methylation of H3K27 while ERα was suppressed. These H3K27 and ERα effects were rescued by simultaneous inhibition of EZH2. Notably, inhibition of JNK failed to have similar effects on ERα expression. However, p38 also regulates the relaxation of chromatin, the phosphorylation of histone H3, and chromatin demethylation via mechanisms that may not linearly exploit EZH2[53–55]. Future studies will clarify the exact mechanisms how p38-dependent stress pathway engages the epigenetic machinery to regulate the ERα expression.

In human PDEC-N and PDEC-BC explants, the stiff LMx-Ag matrix did not induce activation of the stress pathway or ERα expression. Curiously, the genetic profiles of the stiff LMx-Ag cultured human explants were similar to the mouse explants, which were cultured in the soft LMx-Al. When the effective stiffness in the human explant cultures was increased to a level of 400 kPa via magnet-mediated compression, the explants responded by activating the stress pathway and by upregulating the nuclear ERα expression. These findings together with the fact that human mammary epithelial cells naturally encounter higher mechanical stresses than mouse mammary epithelial cells (Supplementary Fig. 9), postulate that human explants require a higher level of stiffness for ERα expression than that exerted by the standard culture matrices.

The present study finds that the matrix elastic modulus (*E*) of 20–30 kPa or higher is required, for the activation of the stress pathway and ERα expression in the mouse explants. However, about 20-fold higher effective stiffness (*E* ≈ 400 kPa) is required for the onset of the stress and hormonal pathways in the human explants. The effective elastic modulus in intact ERα + mouse mammary gland is in the range of 140–200 Pa and in the healthy human ERα + breast around 10–42 kPa (Supplementary Fig. 9, please note that elastic modulus values obtained in rheometric analysis are not directly comparable to metrics obtained via Atomic Force Microscopy (AFM) techniques). Therefore, the mechanical stress needed for ERα expression is clearly higher in the explant cultures than in vivo. This difference may reflect the fact that during the one-week culture period cells soften the surrounding culture matrix through cell secreted proteases and other biomolecules. Hence, the initial stiffness needs to be high enough to provide mechanical stress to the cells until the end of the experiment. However, the difference between in vivo and ex vivo requirements may also arise from other factors present in the complex breast tissue in vivo but lacking in the ex vivo cultures.

In breast tumor tissue sections, nuclear ERα correlated with the level of SAPK/p38 activity and, furthermore, the extent of ERα expression correlated with a higher mammographic density, a known breast cancer risk factor. These data infer a role for mechanical stress in the regulation of ERα in the context of breast tumor and tumor-predisposing conditions. It is well established by number of studies that tumors have a high interstitial pressure and they form a rigid mass representing a significantly stiffer tissue environment than the normal breast epithelium[56]. These compressive forces range around 20-200 kPa. Therefore, in the breast cancer, compressive stress could contribute to the extremely widespread expression of ERα commonly observed in ERα + breast cancers. In our dataset, the number of ERα + cells in the ERα + tumor samples exceeded 75% in 80% of the cases, a much higher figure than that observed in the normal mammary gland with only 10 to 15% ERα + cells[57]. Compressive stress also applies to ERα- breast cancer subtypes such as TNBC and the reasons for the absence of ERα in these cancers remain to be clarified. Curiously, many TNBCs overexpress EZH2 or exhibit dysregulation in other histone methyltransferases and thus, such

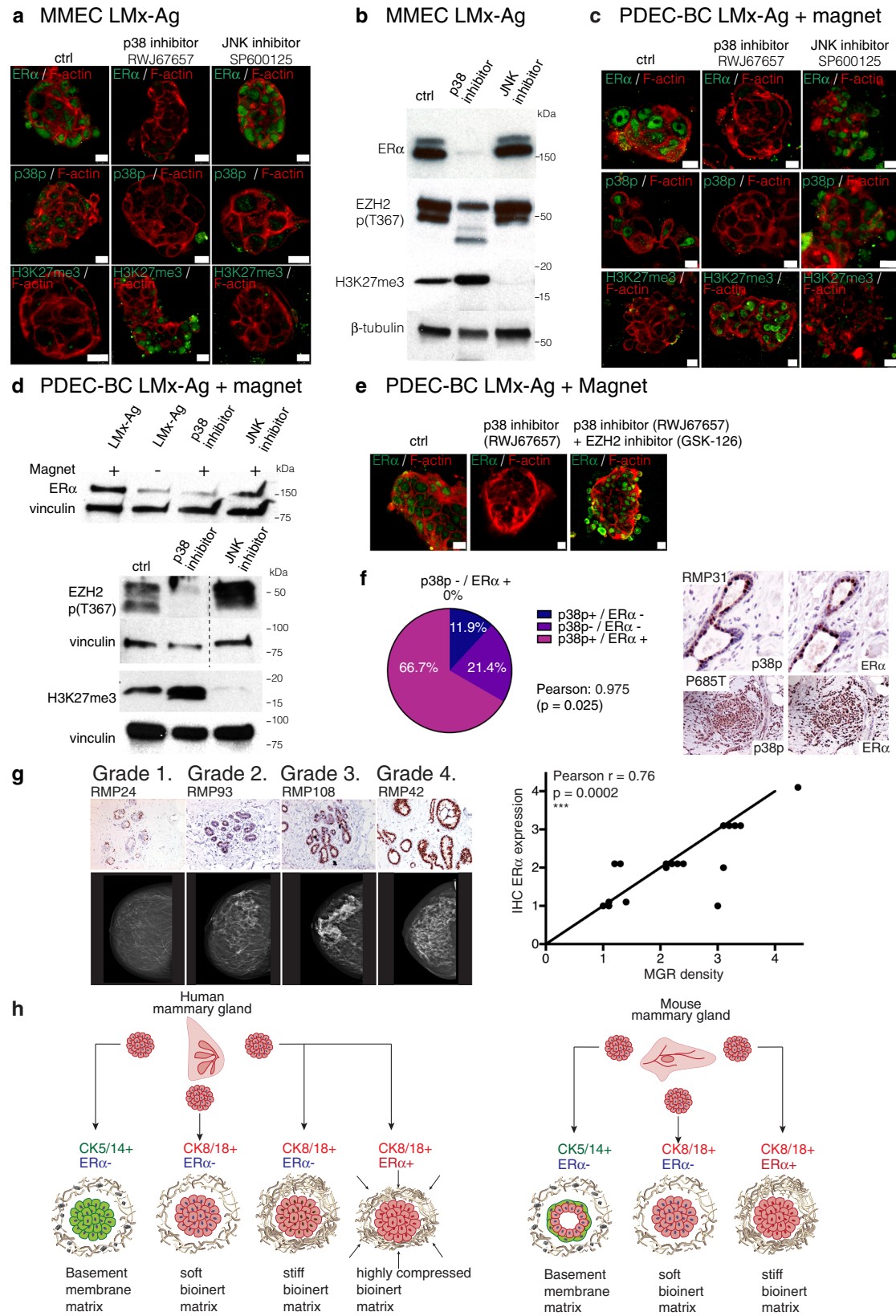

intrinsic cell mechanisms could in some, but likely not in all, cases explain the absence of ERα despite the presence of compressive stress[43,58].

To conclude, our results show that ERα expression is dynamically regulated by matrix-dependent mechanical forces and consequent stress pathway activation. Further downstream, the stress signaling pathway is coupled to EZH2-mediated epigenetic regulation of ERα expression. The surprising cell plasticity and matrix-dependent programmability of differentiated mammary epithelial cells and breast cancer cells in authentic patient-derived tissue cultures open up new opportunities to understand breast cancer biology and to develop new breast cancer treatments.

**Fig. 7 Stress signaling and breast density associates with enhanced ERα expression in women. a** Immunofluorescent staining of p38p, ERα, and H3K27me3 expression in LMx-Ag cultured MMECs after p38 (RWJ67657) and JNK (SP600125) inhibition. N = 4 explants examined from five biologically independent samples. **b** Western blot images of ERα, EZH2 p(T367), and H3K27me3 expression in LMx-Ag cultured MMECs after p38 (RWJ67657) and JNK (SP600125) inhibition (n = 3). **c** Immunofluorescent staining of p38p, ERα, and H3K27me3 expression in magnetic compressed LMx-Ag cultured PDEC-BCs after p38 (RWJ67657) and JNK (SP600125) inhibition. N= 4 explants examined from five biologically independent samples. **d** Western blot images of EZH2 p(T367), H3K27me3, and ERα (with and without magnet) expression in magnet compressed LMx-Ag cultured PDEC-BC after p38 (RWJ67657) and JNK (SP600125) inhibition (n= 3). **e** Immunofluorescent images of magnet compressed PDEC-BCs treated with and without p38 inhibitor (RWJ67657) and EZH2 inhibitor (GSK-126). Expression of ERα is shown in green and F-actin is shown in red. N= 4 explants examined from five biologically independent samples. **f** Immunohistological staining of phosphorylated p38 and ERα in reduction mammoplasty (RMP) and breast cancer samples (PxxxT). The pie chart shows the coincidence of ERα and phospho-p38 expression in breast cancer samples (n = 42). Sequential sections were stained. The statistical significance was calculated between two cohorts (a: n = 17 and b: n = 25) using 2-sided Pearson's correlation test, indicating that the values in the four classifications are linearly correlated between the two data sets. **g**, Correlation between ERα+ positivity and the breast mammographic density (MGR) in reduction mammoplasty samples. Grade 1: samples with 10–24% of ERα positive cells; Grade 2. Samples with 25–49% of ERα+ cells; Grade 3. Samples with 50–74% of ERα+ cells; Grade 4. Samples with 75–100% of ERα+ cells. N = 18 from biologically independent samples. **h** Model figure summarizing the main findings. Scale bar 10 μm.

## Methods

**Cell lines and reagents**. Breast cancer cell lines were purchased from ATCC. DU4475, HCC38, BT-549, and HCC1806 cells were cultured in a RPMI medium (Gibco) supplemented with 10% FBS (Biowest), 2 mM L-Glutamine (Gibco), and 100 U penicillin-streptomycin (Gibco). Additionally, 0.023 IU/mL of insulin (Sigma) was added to BT-549 cells. MCF7 and T47D were cultured in a DMEM medium (Sigma) supplemented with 10% FBS (Biowest), 2 mM L-Glutamine (Gibco), and 100 U penicillin-streptomycin (Gibco). BT-20 cells were cultured in a EMEM medium (Lonza) and MDA-MB-468 cells were cultured in L-15 (Leibowitz's) medium (Lonza), both supplemented with 10% FBS (Biowest), 2 mM L-Glutamine (Gibco), and 100 U penicillin-streptomycin (Gibco). PDEC explants were cultured in MammoCult (StemCell technologies), and the MammoCult media was supplemented with MammoCult proliferation supplement #05622 (StemCell technologies), 4 μg/mL heparin, 20 μg/mL gentamicin (Sigma), 0.1 μg/mL amphotericin B (Biowest) and 10,000 U/mL penicillin/streptomycin (Lonza). Cells were grown in a humidified incubator at 37 °C under 5% $CO^2$, and atmospheric oxygen levels.

Cells were treated with 2 ng/ml human recombinant TGFβ (240-B002/CF, R&D systems), 25 ng/ml – 25 μg/ml of anisomycin (CST), 10 μM EZH2 inhibitor (GSK-126), and 0.5 nM Bortezomib (obtained from the High Throughput Biomedicine unit at the Finnish Institute for Molecular Medicine). Explants were treated with 1 nM 17β-Estradiol (Sigma-Aldrich/Merck), 100 nM 4OH-tamoxifen, 100 nM fulvestrant, 100 nM GDC-0927, 100 nM GNE-274, and 1 μM GDC-0810 overnight.

**Isolation of biological material and three-dimensional (3D) culturing**. Fresh tissue was obtained from the elective breast cancer surgeries performed at the Helsinki University Central Hospital (Ethical permit: 243/13/03/02/2013/ TMK02 157 and HUS/2697/2019 approved by the Helsinki University Hospital Ethical Committee). Patients participated in the study by signing an informed consent form following the Declaration of Helsinki principles. Tissues were collected from reduction mammoplasty samples, from tumors, and the adjacent normal-like areas of the tumors. From each tumor and normal tissue specimen, a portion was taken for immunohistochemical and DNA/RNA/protein analysis, and the reminder was used for the 3D cultures. Explants were produced by incubating the samples overnight in collagenase A (1–3 mg/ml; Sigma) containing the MammoCult media (StemCell technologies) with gentle shaking (130 rpm) at +37 °C. The resulting explants were collected via centrifugation at 353 rcf for 3 min and washed once with the culture medium. Isolated explants were embedded in different 3D matrices and plated on 8-chamber slides (Thermo Scientific). Those matrices used for the cultures were: Matrigel (growth factor reduced, Corning), GrowDex (UPM), Collagen I Rat tail (Corning), egg white, ovomucin isolated from the egg white, alginate (Alginic acid sodium salt W201502, Aldrich, lot: MKBV5260V), alginate + RGD (Sodium Alginate MVG GRGDSP, NovaMatrix, NOVATACH, lot: BP-1730-06, BP-1802-04), and agarose (UltraPure™ Low Melting Point Agarose, Invitrogen).

MMECs were isolated through a 1 h collagenase A treatment with gentle rocking (140 rpm) at +37 °C. The media used for the mouse tissue was DMEM/F12 (Thermo Fisher Scientific) supplemented with 5% horse serum (Life technologies), 5 ug/mL Insulin (Sigma), 1 μg/mL hydrocortisone (Sigma), 10 ng/mL EGF (Sigma), and 100 U PenStrep (Lonza). Four month old female NMRI mice were maintained in a pathogen-free (SPF) facility at the University of Helsinki. The mice were maintained under standard conditions in ventilated animal cages at +21-23 °C, 50–70% humidity, 12 h dark and light cycle with standard diet and water. The mice used in the study were sacrificed with $CO_2$ followed by cervical dislocation. All animals were covered by a license (ESAVI-2010-05551_Ym-23, KEK19-002) approved by the National Animal Experiment Board of Finland (Eläinkoelautakunta, ELLA).

**Preparation of 3D matrices**. Agarose solutions were prepared by first dispersing 0.07 g of the UltraPure™ Low Melting Point Agarose (Invitrogen, 16520050) in

1 mL of 1xPBS followed by heating the mixtures until the agarose was completely dissolved at 80 °C for 20 min. Alginate gels were prepared by dispersing 0.07 g of alginic acid sodium salt or sodium alginate powder (Sigma-Aldrich/Merck W201502) in 1 mL of 1 x PBS and heating the mixtures until the alginate was completely dissolved at 80 °C for 20 min. After melting the gels, the temperature was decreased down to 40 °C and 10 μl of explants was mixed with 40 ul/well of the pre-cooled gel on an 8-well chamber slide. 0.5 mL of Mammocult media per 8-chamber slide well was added after the matrix was solidified. GrowDex was a pre-prepared commercial matrix. Matrigel 8.8 mg/mL was used as is or mixed with a pre-cooled DMEM to obtain the desired concentration. Solutions were kept at 37 °C for 30 min to form a gel. Collagen solutions were prepared by partly following the Alternate Gelation Procedure for BD™ Collagen I, rat tail. All the components and equipment were pre-cooled and kept on ice during the preparation. First,10 x PBS and water were mixed. Then, a Collagen I rat tail stock solution was added. Finally, 1 M NaOH was added and mixed throughout. Solutions were kept at 37 °C for 30 min to form a gel. Egg whites were separated from the yolks and filtered through a sinter to keep only the clear part. Egg whites were incubated at 60 °C for 1 h to form a gel. Ovomucin gels were prepared according to an established protocol[59].

**Protein extraction from LMx-Ag cultures**. Protein extraction was performed using Precellys 24 Homogenizer (#P000669-PR240-A) using Precellys Soft Tissue Lysing Kit CK14 (catalog #P000912-LYSK0) and Precellys Hard Tissue Lysing Kit CK28 (catalog #P000911-LYSK0-A). 3D culture specimens were first transferred into Precellys Soft Tissue Lysing Kit. Lysis Buffer (50 mM TrisHCL pH 7.4; 1% SDS; 5.5% Triton X-100 in MQ-H2O) was added and homogenized 3 × 20 sec at 5.500 rpm with 2 min cooling period in between each homogenization. Following, the liquid was pipetted from the bottom of the tube (below beads) into a fresh 1.5 mL reaction tube which was then spun down briefly using a table centrifuge. Afterwards, the supernatant was collected into a fresh 1.5 mL reaction tube. The beads from Precellys Hard Tissue Lysing Kit were added into the already used Precellys Soft Tissue Lysing Kit tube. After adding Lysis Buffer again, the same extraction was performed. The lysate from the second extraction was collected from the top of the tube (above beads) and collected into separate 1.5 mL reaction tubes.

**Scanning electron microscopy (SEM) imaging**. The specimen for SEM imaging were prepared using different sample preparation methods to obtain aerogels. GrowDex and agarose samples were plunge-freezed in liquid propane and subsequently lyophilized. The alginate sample was prepared using a supercritical carbon dioxide drying method. Matrigel, ovomucin, collagen and alginate-RGD samples were prepared using glutaraldehyde (3.5%) fixation for overnight. After washing the excess fixative with water, the specimen was dehydrated with a series of water/ethanol (70/30, 50/50, 30/70, 10/90 and 0/100 v/v) mixture by incubating 10 min in each solution. Finally, samples were chemically dried with hexamethyldisilazane (HMDS) by incubating in each solution for 10 min in 1:1 (v/v) ethanol: HMDS and 2 x HMDS.

SEM images were acquired with a Zeiss Sigma VP scanning electron microscope with an acceleration voltage of 1.0 to 1.5 kV. The aerogel sample was placed on the carbon tape and sputter-coated with Au/Pd with Emitech K950X/K350 or a Leica EM ACE600 high vacuum sputter coater. Sample preparation methods, coatings, and acceleration voltages for each matrix appear in Supplementary Table 1a.

**Rheology**. We carried out oscillatory rheology using a TA Instrument AR2000 stress-controlled rheometer with a Peltier heated plate. 20 mm steel parallel-plate geometry and 20 mm cross-hatched steel parallel plate geometry were used depending on the sample. The sealing lid and silicon oil prevented evaporation during the measurement. The gap temperature compensation parameter was 0.7 μm/°C. First, the linear

viscoelastic region (LVR) was confirmed with a preliminary measurement including strain sweep. Secondly, time sweeps were used to follow the gelation process as well to establish the stability of the gels at 37 °C. Thirdly, the frequency and strain sweeps were measured to compare the rheological properties of different matrices at 37 °C. The parameters for the measurements appear in Supplementary Table 1b for the specific matrices. Data were acquired in triplicate and reported as average unless otherwise stated. The storage modulus, $G'$, the loss modulus $G''$, the absolute shear modulus $|G^*|$ and the phase angle are obtained from the dynamic oscillation. $G'$ indicates the matrix elasticity, whereas $G''$ relates to the viscous losses. $|G^*|$ describes the overall shear modulus and it is the absolute value of the complex modulus $(|G^*| = \sqrt{G'^2 + G''^2})$.

**Immunofluorescent staining.** Three-dimensional (3D) cultured breast cancer explants were fixed with 4% paraformaldehyde for 15 min at room temperature and washed three times with PBS. The tissue explants were permeabilized with 0.5% Triton X-100 in PBS for 10 min at RT and blocked in an IF buffer (0.1% BSA, 0.2% Triton X-100, 7.7 mM $NaN_3$, and 0.05% Tween 20 in PBS) supplemented with 10% (v/v) normal goat serum for 1 h. Explants were then incubated with the primary antibody diluted in a blocking solution overnight at 4 °C. Following incubation, explants were washed three times with an IF buffer and then incubated using the appropriate Alexa Fluor secondary antibody diluted in an IF buffer with 10% goat serum. The list of used antibodies are shown in Supplementary Data 2. After 60 min of incubation at RT, the explants were washed with an IF buffer as before and the nuclei were counterstained with Hoechst 33258 (Sigma). Slides containing tissue explants were mounted with the ImmuMount reagent (Fisher Scientific). Images of the structures were acquired using a Leica TCS SP8 CARS confocal microscope using an HC PL APO CS2 40x objective (Biomedicum Imaging Unit, University of Helsinki).

**Immunohistochemistry.** Tissues and explant cultures were fixed with 4% paraformaldehyde (PFA) and embedded in paraffin. The samples were sectioned in 5 µm slices and deparaffinized. The heat-induced antigen retrieval was performed whether with a microwave oven or a pressure cooker in a citrate buffer solution (Dako). Histochemical stainings were carried out using standard techniques for IHC and IHC-IF. Images were taken with a Leica DM LB microscope or with a Zeiss AxioImager 1 (Biomedicum Imaging Unit, University of Helsinki). The list of used antibodies are shown in Supplementary Data 2.

**DNA/RNA sequencing and data analysis.** Total RNA was isolated using RNeasy (Qiagen) or Trizol (Thermo Fisher), and the DNAase removal step was performed after the isolation (Zymo research). RNA sequencing libraries were prepared from 100 ng of total RNA using either the ScriptSeq Complete Gold Kit or the NEBNext Ultra Directional RNA Library Prep Kit for Illumina depending on the RNA integrity. Using the ScriptSeq Complete Gold Kit, the ribosomal RNA was removed first from the total RNA using the Ribo-Zero™ Gold rRNA Removal Kit after which the RNA was fragmented chemically. The libraries were prepared according to the manufacturer's instructions. Finally, the library was assessed with the Agilent Bioanalyzer.

The NEBNext Ultra Directional RNA Library Prep Kit for Illumina was used to generate the cDNA libraries for next generation sequencing. First, the ribosomal RNA depleted samples (10 ng) were fragmented to generate the inserts around 200 bp. The libraries were prepared according to the manufacturer's instructions. The library quality was assessed with Bioanalyzer (Agilent DNA High Sensitivity chip) and the library quantity with the Qubit (Invitrogen).

Samples were sequenced with the NextSeq 500—Illumina instrument using 75 PE reads with a sequencing depth of 33 M reads/sample. Differentially expressed genes between different groups were found using state-of-the-art statistical methods and packages, such as edgeR/DESeq2. The Gene Set Enrichment Analysis 3.0 (Broad Institute) was used to analyze the differences in the gene expression profiles[60]. GSEA results were visualized using Cytoscape (v.3.7.2) and the enrichment map plug-in[61].

DNA was extracted from the original tumors and the corresponding 3D cultured samples. The DNA integrity was confirmed using gel electrophoresis. The TruSeq Amplicon Cancer Panel (TSACP, Illumina), which covered the hotspot regions of 48 genes, was selected for the mutational profiling. The sequencing libraries were performed according to the manufacturer's instructions and the samples were sequenced with the MiSeq sequencer (Illumina). The MiSeq reporter was used for the data analysis and the GATK tool was used for variant calling.

The BRB-sequencing method was based on the Drop-seq protocol described in Macosko EZ et al. Cell 2015[62]. First the RNA samples (10 ng) were barcoded using Indexing Oligonucleotides (Integrated DNA Technologies). cDNA was prepared from RNA samples (10 ng) using RT mix containing Maxima RT buffer, 1 mM dNTPs, Maxima H-RTase (all ThermoFisher Scientific) and Template Switch Oligo (Integrated DNA Technologies). RiboLock (ThermoFisher Scientific) was used to inhibit the RNases. The samples were incubated in T100 thermal cycler (BioRad).

The cDNA was amplified by PCR using RT mix as template, 1x HiFi HotStart Readymix (Kapa Biosystems) and SMART PCR primer. The samples were thermocycled in a T100 thermocycler (BioRad). The PCR products were pooled together in sets of 12 samples containing different Indexing Oligos and purified with 0.6X Agencourt AMPure XP Beads (Beckman Coulter) according to the manufacturer's instructions. The purified cDNA was tagmented using the Nextera kit. The reaction was performed according to manufacturer's instructions, apart from the P5 SMART primer that was used instead of S5xx Nextera primer.

The concentration of the libraries was measured using a Qubit 2 fluorometer (Invitrogen) and the Qubit DNA HS Assay Kit (ThermoFisher Scientific). The quality of the sequencing libraries was assessed using the LabChip GXII Touch HT (PerkinElmer), with the DNA High Sensitivity Assay (PerkinElmer) and the DNA 5 K / RNA / Charge Variant Assay LabChip (PerkinElmer). The libraries were sequenced on a Illumina NextSeq 500, with a custom primer producing read 1 of 20 bp and read 2 (paired end) of 50 bp. Sequencing was performed at the Functional Genomics Unit of the University of Helsinki, Finland. Primer sequences are shown in the Supplementary Table 2.

**qRT-PCR analyses.** Total RNA was isolated from cell lines and primary cell cultures using the Qiagen RNEasy Kit according to the manufacturer's instructions, while the cDNA synthesis was performed with the Maxima First Strand cDNA Synthesis Kit for RT-qPCR (Thermo Scientific). Real-time RT-PCR was performed with LightCycler® 480 II (Roche) using DyNAmo ColorFlash SYBR Green (Thermo Scientific). The gene-specific primer sets were used at a final concentration of 0.2 mM to detect changes in PGR, and GREB1. All qRT-PCR assays were performed in at least three biological replicates. Primer sequences for the human samples were taken from Kangaspeska et al. 2016[63]. The primer sequences for mice samples are shown in Supplementary Table 2.

The relative gene expression was analyzed using the Livak-Schmittgen $(2^{-\Delta\Delta Cq})$ method[64].

**Magnetic force-mediated compression.** The explants were embedded within LMx-Ag. The explant cultures had a round shape with a radius of $R_{matrix} = 2.5$ mm and an initial thickness of $l_0 = 2.0$ mm. Following preparation, two magnets (Magnet Expert Ltd; #F643-SC) were used to compress the cultures between the well-plate bottom and a metallic grid on the top of the cell culture. A vertical compressive force of $F = 0.724$ N was applied on each culture by the magnets (0.711 N; calculated based on Abbott et al., 2007)[65] and the top magnet/grid weight (0.013 N). Compression ($\sigma_{compression}$) by the magnets is defined by: $\sigma_{compression} = \frac{F}{A}$, where $A$ is the cross-sectional area of each explant culture. The explant cultures were compressed by $\sigma_{compression} = 37$ kPa. Specifically, each explant culture with an initial volume of $V_0 = l_0 \pi R_{matrix}^2$ was compressed volumetrically by $\Delta V$. Therefore, the vertical strain $\varepsilon_{compression}$ related to the compression was estimated based on $\sigma_{compression}$ and the bulk modulus ($K$) definition: $\varepsilon_{compression} \approx \frac{\Delta V}{V_0} = \frac{\sigma_{compression}}{K}$, where $\Delta V \approx \Delta l \pi R_{matrix}^2$ and $\Delta l$ is the vertical compression-related thickness change. Furthermore, K is defined using the initial absolute shear modulus $|G^*|$ of the explant matrix: $K = \frac{2(1+\upsilon)|G^*|}{3(1-2\upsilon)}$, where $\upsilon$ is the Poisson's ratio of agarose: $\upsilon = 0.44$ is based on[66]. The, the effective elastic modulus of the compressed explant-culture matrix is $E_{effective} = \frac{\sigma_{compression}}{\varepsilon_{compression}}$. The effective shear modulus $|G^*|_{effective} = \frac{E_{effective}}{2(1+\upsilon)}$. In particular, compressing the explant culture involved $|G^*|_{effective} = 129$ kPa that corresponds to a proportional and an absolute increase of 178% and 83 kPa to the initial average $|G^*|$ value, respectively. In addition, compressing involved an increase of the average vertical stresses within the explant culture by $\sigma_{compression} = 37$ kPa. To summarize, compressing alters the order of magnitude for both effective moduli $E_{effective}$ and $|G^*|_{effective}$ as well as for the internal-stress properties.

**Statistics and reproducibility.** We report our results as the mean ± standard deviation (SD). Datasets were analyzed using unpaired Student's t-tests or the Mann-Whitney U test. All the experiments with representative images (Western blot, immunohistology, and immunofluorescence stainings) have been repeated at least thrice. When comparing multiple groups, the p values were calculated using ANOVA. Dunnett's multiple comparisons post hoc test was used with ANOVA. Statistical analyses were performed using Graphpad Prism 8 (Version 8.4.3).

**Western blotting.** Whole-cell extracts were isolated using a RIPA lysis buffer supplemented with protease (Roche) and phosphatase inhibitors (Roche). The nuclei were broken with a 20 G needle. The concentration of the isolated proteins was determined using the BCA Protein Assay Reagent (BioRad). Fifteen to twenty micrograms of protein were separated on the BioRad gradient gels at 4% to 20% and transferred to nitrocellulose membranes (BioRad). The membranes were then incubated with the appropriate primary and secondary antibodies according to the manufacturer's recommendations. The list of used antibodies are shown in Supplementary Data 2.

**Reporting Summary.** Further information on research design is available in the Nature Research Reporting Summary linked to this article.

## Data availability
The RNA sequencing data generated in this study have been deposited in Sequence Read Archive (SRA) database and are accessible through the SRA accession numbers:

PRJNA663587, PRJNA663448, PRJNA663028. The BRB-RNA sequencing data generated in this study have been deposited in Sequence Read Archive (SRA) database and are accessible through the SRA accession numbers: PRJNA775661 and PRJNA775657. Due to the nature of the consent given by the patients, we are not allowed to share the exome sequencing data in any public data repositories. The publicly available H3K27me3 ChIP-seq data from MDA-MB-231, MDA-MB-453, and SUM-159PT cell lines are available in NCBI GEO database under accession code GSE38548. The publicly available H3K27me3 ChIP-seq data from MDA-MB-436 cell line are available in NCBI GEO database under accession code GSE62907. The publicly available H3K27me3 ChIP-seq data from MCF7 cell line are available in ENCODE database under accession code ENCSR761DLU_2. The publicly available H3K27me3 ChIP-seq data from UACC812 and ZR-75-1 cell lines are available in NCBI GEO database under accession code GSE85158. The publicly available H3K27me3 ChIP-seq data from T47D cell line are available in NCBI GEO database under accession code GSE63109. The publicly available H3K27me3 ChIP-seq data from ZR-75-30 cell line are available in NCBI GEO database under accession code GSE71323. Source data are provided with this paper. The remaining data are available within the Article, Supplementary Information or Source Data file. Source data are provided with this paper.

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

## Acknowledgements

We are grateful to the patients who participated in this research and made it possible, and to the surgical personnel at Helsinki University Hospital who assisted with the recruitment and collection of the sample material. We thank Leena Saikko, Tiina Raatikainen, and Maiju Merisalo-Soikkeli for their excellent technical support. We thank Vanessa Fuller for the comments on the manuscript and Juhi Somani for assistance with statistical analysis. We thank the Biomedicum Functional Genomics Unit (FuGU) for their high-quality genome profiling services, the Biomedicum Imaging Unit (BIU) for the microscopy support, and the Laboratory Animal Center (LAC) of the University of Helsinki for providing the mice used in this work. We acknowledge the provision of facilities and technical support by Aalto University at OtaNano—Nanomicroscopy Center (Aalto-NMC). This study was funded by grants from the IMI EU-EFPIA PRE-DECT 115188, Business Finland (Grant No: 544/31/2015 and 2489/31/2017), the Academy of Finland, Academy of Finland Centre of Excellence (HYBER 2014-2019) and iCAN Flagship, ERC-Advanced Grant (DRIVEN), the Finnish Cancer Organization, the Sigrid Juselius Foundation, a Biocentrum Helsinki collaboration grant, Finnish Cancer Institute (FCI), HiLIFE, Jane and Aatos Erkko Foundation, and RESCUER project, which has received funding from the European Union's Horizon 2020 research and innovation programme under grant agreement No. 847912. L.M. acknowledges The Finnish Foundation for Technology Promotion, Walter Ahlström Foundation and Jenny and Antti Wihuri Foundation for financial support. J. P. acknowledges the Instrumentarium Science Foundation grant of an Instrufoundation fellow.

## Author contributions

Study design: P.M.M., L.M., K.B., N.N., H.A.-H., M.K., C. M., M.R.J., L.N., I.R., A.P., J.R., M.S., B.H., L.E., K.Y., B.S., L.P., J.K., O.I.; writing group; P.M.M., L.M., K.B., N.N., I.R., H.J., J.P., J.P., O.M., J.K., O.I. data analysis: P.M.M., L.M., K.B., N.N., H.A.-H., M.K., M.M., P.K., J.R., J.P., L.N., I.R., A.P., K.H.; bioinformatic analysis: M.K., T.S., L.E., P.M.M., A.P. J.V.; provided patient material and clinical data/analysis: M.L., P.H., J.M., H.J., P.K., T.M., M.M., K.H. All authors read and approved the final version of the paper.

## Competing interests

The authors declare no competing interests.
