## [Peer Review File · Nature Communications]

Compressive Stress-Mediated p38 Activation Required for ER α + Phenotype in Breast CancerReviewers' Comments:

Reviewer #1:

Remarks to the Author:

This is an interesting study demonstrating that the stiffness of the matrix regulates expression of ERα in normal breast tissue and in luminal breast cancers. The most compelling data are those related to the patient-derived tumor explants, and the experiments showing that increasing the stiffness of the matrix increases ERα expression.

However, the data that links matrix stiffness with p38 phosphorylation, H3K27me3, and ERα needs significantly more mechanistic approaches. At this time, the link between these factors is associative. For example, the authors do not show whether inhibition of p38 phosphorylation prevents the effect of matrix stiffness on H3K27me3 and on ERα expression. The effect of p38 phosphorylation on H3K27me3 binding ERα using ChIP assays are not investigated.

The other aspect is that some of the data presented are not novel, and are correlative. The association between high EZH2 and negative ER expression has been reported in 2003 (Kleer et al), and later corroborated by several studies. The association shown between phospho p38 and ER in the 17 invasive carcinomas is preliminary. Additional cases are needed to conclude a statistically significant association in cancer. Data in Fig. 7 is correlative. The association between mammographic density and ER expression is interesting. However data are shown in normal tissues, and not in cancer. Further study in cancer is warranted.

Reviewer #2:

The present manuscript from Munne et al describes experiments that aim to establish an *ex vivo* luminal ER α + breast cancer model with patient-derived breast explants to restore ER α expression and study ER α + signalling pathway.

Using MMECs and an impressive number of human breast tissue and breast tumor samples, the authors show that maintaining luminal epithelial phenotypes for both normal breast and breast cancer explants *ex vivo* is not cell identity-dependent, but matrix-dependent. They distinguish between matrix scaffolds which are luminal preserving matrices (LMx) and basal promoting matrices (BMx). The transcriptomic profiles and ER α expressions in MMEC are matrix stiffness-dependent and mediated by stress and H3K27me3 pathways. The authors demonstrate that about 20-fold higher effective stiffness is required to activate stress and hormonal pathways in the human explants than in the mouse explants.

In summary, matrix stiffness induces p38 stress pathway, represses EZH2-dependent H3K27me3, and upregulates ER α expression. These findings if fully supported by data are novel and will be of interest to the field.

Major Concerns:

1. The main concern about this manuscript is whether the stiffness of agarose matches the stiffness of human breast carcinoma, which determines the biological/clinical relevance of this *ex vivo* culture model.

From many works of Valerie M. Weaver's, the human breast carcinoma has a stiffness of around 2 kPa measured by AFM, which is much lower than the effective elastic modulus (indicating stiffness) of agarose+compression (373kPa) used in this manuscript. Could authors explain where the difference of stiffness comes from? Is there a dependency with the techniques used (the authors used rheological test instead of AFM)?

Acerbi, Irene, ..., Valerie M. Weaver. *Integrative Biology* (2015)

The authors referred matrix stiffness to different terminologies, e.g. storage modulus, elastic modulus and complex modulus, which is difficult and confusing for readers without professional Materials background. Could authors find a way to make it simpler and clearer? For instance, would it work that the authors specify the relationship between each terminology and use one of them to represent stiffness consistently?

2. The authors showed that epithelial cell identity was not a stable feature in a culture but highly sensitive to changes mediated by the matrix environment, and they demonstrated this by comparing the stiffness differences mainly between alginate (soft) and agarose (stiff), as well as in agarose with different stiffness. However, are there other aspects, such as structural and/or chemical property differences among alginate, agarose and other luminal preserving materials tested (egg white and ovomucin), that needs to be considered? For instance, if the authors could acquire

stiffness gradients in other materials than merely in agarose by changing the material concentration or applying the magnet-mediated compression that they used to further increase the stiffness of agarose, is it possible that ER expression and function would also be restored?

Minor Concerns:

3. In line 361: the authors claim a **strong** positive correlation between p38 and ER α protein expression levels. However, the R^2 value is only 0.12, and the Spearman and Pearson correlation coefficient are around 0.3.
4. Have the authors tried to induce ER α expression by anisomycin with other TNBC cell lines than DU4475 cell line?
5. The authors chemically induced stress by anisomycin and this successfully upregulated the ER protein expression in TNBC cell line. Have the authors tried to mechanically induce stress and ER α expression in TNBC cell lines or TNBC patient-derived breast cancer explants by magnet-mediated compression?
6. Anisomycin is a potent activator of stress-activated protein kinases (JNK/SAPK) and p38 MAP kinase. Acts as a potent signaling agonist to selectively elicit homologous desensitization of immediate early gene induction (c-fos, fosB, c-jun, junB and junD). How they can rule out the inhibition of JNK as equally important for ER?
7. In Methods, the authors did not specify the experimental process of tissue embedment in matrix in either "Isolation of Biological Material and Three-Dimensional (3D) Culturing" or "Preparation of 3D Matrices" section, is it possible for the authors to provide experiment procedures and details so that other researchers can reproduce the experiment?
8. The authors did not provide the images of CTRL conditions serving as references or base lines for comparison in some figures. For instance, in figure 2a and 2b, it would be nice to have immunofluorescence images of MMEC and PDEC-N that were not cultured in any of the matrices to provide naïve expression levels of luminal and basal markers. It is also helpful if authors specify the sample code in every figure, for example, in figure 1c, the authors only marked PDEC-BC with a sample code, but did not mark PDEC-N.
9. In figure 2a, the authors did not show images of tissue embedded in ovomucin even though they mentioned this in the figure legend and in the main text.
10. In line 266 "the ER α -regulated gene **sets** were clearly diminished in the treated samples (Fig. 4I)", there is only one gene PGR in figure 4I, no gene sets. In figure S3e, there is another gene GREB1. The author need to either put two gene expression plots together, or mention the supplementary figure in the brackets at the end of this sentence.
11. In line 286, the authors mentioned the PDEC-N was shown in figure 4 and figure S5, but there is no enrichment map of PDEC-N in figure S5.
12. In figure S2, the author did not specify the difference between figure S2 m and S2 n with egg white.
13. Lines 88-90: authors do not mention PDXs. There are few other models that can retain functional ER signaling (e.g PMID: 26947176).

14. Line 103-104. Authors should mention recent study of 2% alginate (PMID: 32807212).
15. Make sure the contexts *in vivo* and *ex vivo* are all in Italic in the paper.
16. In the legend of Fig 2, there is a typo in the function $E = 2(1-\nu)G^*$. The correct one should be $E = 2(1+\nu)G^*$.
17. Typo in line 661: there are two “where”, delete one.
18. Typo in legend of figure S3f, 24 may be 24 h.
19. Fig. 4i: to write $ER\alpha$ and not ERa
20. Authors they should report more comprehensively in the methods how to construct agarose extracellular matrix scaffold.

Reviewer #3:

Remarks to the Author:

In this study, Munne et al analyze the regulation of estrogen receptor (ERa) in normal and transformed mammary epithelial cells. They use 3D cultures under distinct conditions to demonstrate that matrix density and stiffness, stress signaling (p38) and reduced EZH2 mediated histone methylation are all important to sustain ERa expression in these cells. Notably, using human breast tissue from mammoplasty, they show a correlation between active p38 stress signaling and ERa expression. Moreover, they show that breast tissue density was associate with increased ERa expression. There are several interesting aspects to this study. It proposes microenvironmental regulation of ERa expression in mammary cells and this may be highly consequential. However, there are a number of important caveats and loose ends that need attention. The study is rather fragmented and therefore it is difficult to get a clear overall picture of what is going on. Details below.

Major points

1. The authors demonstrate that tissue density and stiffness, stress signaling and EZH2 inhibition all can promote ERa expression. However, the link between the three functions and which stress signaling pathway is involved are not demonstrated clearly and thus still rather speculative. For example, what is shown in Figure 5d is not sufficient to demonstrate a link between tissue stiffness and p38 signaling. First, the comparisons are confusing. Why are the same comparisons not used for MMEC and the PDECs? What type of stress signature is used? Is it specific p38 induced signature or does it include JNK induced genes? This should be clarified. Moreover, to confirm a link between ECM stiffness and p38 signaling, a p-p38 Western blot should be performed on cells in matrices of different stiffness. It is also important to note that anisomycin, that the investigators use to promote stress signaling is not specific for p38 but can also induce JNK signaling. Which stress pathway is required for ERa expression. To sort this out, ERa expression should be analyzed in MMEC grown in LMx-Ag and treated with JNK or p38 inhibitors. To confirm the connection of stiff matrix and stress to EZH2 repression, T367 phosphorylation of EZH2 should be analyzed under the same conditions (LMx-Ag with stress signaling inhibitors). These experiments could also be done with PDEC-N and PDEC-BC in compressed LMx-Ag matrix and with the stress signaling inhibitors.
2. I have several concerns about the phenotypic consequences of ERa activity. The data suggest that there may be a discordance between ERa and luminal phenotype (based on CK8 expression). Whereas LMx-AI, LMx-Ew and LMx-Ag all promote expression of CK8, only LMx-Ag promotes induction of ERa in MMEC and none of the matrices induce ERa in human PDEC samples. Would ERa signaling not be a key regulator of luminal fate in the mammary gland? This needs to be analyzed and explained much better. First, by using luminal gene signature rather than a single marker. In figure 3, only gene signatures that are induced in basal cells are used when comparing BMx and LMx conditions. How proficient are the LMx matrices in promoting luminal phenotype and is there a difference between LMx-Ag and the others? This should be addressed using specific luminal signatures.
3. In Figure 6e, the investigators propose a model where ERa is a direct target of p38 signaling via EZH2. However, they only show that ERa protein is affected by stress signaling. ERa stability is tightly regulated and increased stability can be observed in breast cancer cells. Is the ERa mRNA changed?
4. Figure 5e shows that the luminal breast cancer cells MCF7 and T47D have active p38 signaling. How do they maintain ERa expression under normal conditions and without a specific stiff matrix? Would the results suggest that luminal breast cancer have higher p38 signaling compared to basal breast cancer? This should be looked at.
5. An interesting aspect of the study is that it shows ERa regulation that is highly context dependent. This may have impact on cellular phenotypes and possibly breast cancer subtypes. Specification of basal like breast cancer has been shown to be regulated by the microenvironment (Roswall P et al Nat Med 2018). A discussion of this should be included.

6. In Figure 4j, Ki67% is shown to decrease in response to E2 treatment, even though ERa targets are upregulated with the same treatment. This is surprising and should be explained.

7. In Figure 4i, quantified ER levels are shown with a scatter plot. However, only one condition (LMx-Ag 70mg/ml) is shown. The results from all polymer concentrations should be included in the scatter plot.

8. I think that correlation analyses, that include mRNA or total protein levels of p38, MAP3K1 or other proteins regulated by phosphorylation, do have a rather limited value.

Minor points

1. In figure 3h, MCF7 does not cluster with the others, even under LMx conditions. Why not?

2. Figure legends need more details to explain each panel. There is no need to explain the results, only what each panel contains (how samples were treated and analyzed and to clarify abbreviations).

3. Which post-hoc tests were used with ANOVA in multiple comparisons? This should be described in figure legends or in methods.

4. In Figure 6g, it is difficult to see p-p38 induction in the PDEC-BC sample. Is this correct?

5. There are some minor issues with the Supplementary Figures. Suppl. Figure 3g, and Suppl. Figure 4 are not referred to in the main text. In addition, Suppl. Figure 8 is only one panel, but it is in the text referred to as Suppl. Figure 8a and 8b.

Reviewer #1

This is an interesting study demonstrating that the stiffness of the matrix regulates expression of ER α in normal breast tissue and in luminal breast cancers. The most compelling data are those related to the patient-derived tumor explants, and the experiments showing that increasing the stiffness of the matrix increases ER α expression.

However, the data that links matrix stiffness with p38 phosphorylation, H3K27me3, and ER α needs significantly more mechanistic approaches. At this time, the link between these factors is associative. For example, the authors do not show whether inhibition of p38 phosphorylation prevents the effect of matrix stiffness on H3K27me3 and on ER α expression.

Authors' response (R.1.1): For the revised manuscript, we have performed a substantial amount of new experiments to strengthen the causal roles of p38 and H3K27me3 in ER α regulation. We show that p38 inhibitors suppress ER α and increases H3K27me3 in the magnetic compressed PDEC-BC and in LMx-Ag cultured MMECs. We also show *via* gene expression profiling that the ER α regulated gene sets are suppressed after p38 inhibitor treatment.

In the revision, we have performed additional experiments with two different p38 MAPK inhibitors (RWJ67657, SB203580) and the new results consistently show that inhibition of p38 activity suppresses ER α expression and activity in the stiff matrix conditions (MMEC in LMx-Ag, PDEC-BC in the magnet compressed LMx-Ag).

The new data are added in the revised manuscript as:

- FIGURES: Figure 7 a-d; Supplementary Fig. 6 a, b, d
- TEXT in the RESULTS section:

”To test the functional importance of p38 mediated stress pathway for ER α expression, we chemically inhibited p38 in LMx-Ag cultured MMECs and magnet compressed PDEC-BCs (for validation of p38 MAPK inhibitors, see supplementary Fig. 6 a, b, d). As evidenced by the western blot, RNA sequencing, and immunofluorescence microscopy analysis, inhibition of p38 abolished nuclear ER α expression (Figure 7 a-d) and suppressed ER α activity in both MMEC and PDEC-BC (Supplementary Fig. 6 b, d).”

Furthermore, the inhibition of p38 in the same experiments also suppressed phosphorylation of EZH2-p(T367) and resulted in enhanced trimethylation of H3K27

The new data are added in the revised manuscript as:

- FIGURES: Figure 7 b, d
- TEXT in the RESULTS section:

”In addition, consistent with our earlier notion suggesting involvement of p38 as a mediator of the matrix stiffness to EZH2-mediated trimethylation of H3K27 and downmodulation of ER α activity, inhibition of p38 also suppressed phosphorylation of EZH2-p(T367) and resulted in enhanced trimethylation of H3K27 (Figure 7 b, d). Since phosphorylation of EZH2 at T367 suppresses its activity, our results from both mouse and human explant cultures altogether are consistent with a mechanistic model presented in Fig 6 e. Accordingly, specifically a stiff matrix induces a p38 mediated stress pathway, which keeps EZH2 phosphorylated at T367 thus suppressing the activity of this key enzyme that catalyzes the

addition of methyl groups to histone H3 at lysine 27. In the absence of epigenetic repression, the expression of ER α is favored.”

- DISCUSSION:

”In support, we show that the inhibition of p38 simultaneously prevented the phosphorylation of EZH2 and led to an increase in methylation of H3K27 while ER α was suppressed. This effect was specific to p38 as inhibition of JNK did not have similar effects on ER α expression.”

The new results clearly demonstrate a requirement for p38 in regulation of H3K27me3 and ER α in the mouse and human mammary epithelial cultures and in breast cancer samples.

It is of note that for the new experiments, we needed to establish a completely new method for extracting proteins from agarose gels, which is now described in the:

- METHODS section: “Protein Extraction from LMx-Ag cultures”.

The effect of p38 phosphorylation on H3K27me3 binding ER α using ChIP assays are not investigated.

Authors’ response (R.1.2): We were not able to harvest cells from agarose matrix by lysing the gel with a buffer in a manner that would release the chromatins. This makes the ChIP analysis technically impossible. Therefore, we addressed the connection between phospho-p38 and H3K27me3 in ER α regulation with p38 inhibitors, showing that p38 suppression prevents phosphorylation of EZH2 p(T367), leading to increased levels of H3k27me3 and downregulation of ER α both in the MMEC and PDEC samples. In the revised manuscript, we show with western blot analysis that in stiff matrices, inhibition of p38 suppresses EZH2-p(T367) in both the MMEC and PDEC-BC cultures. Since phosphorylation of EZH2 at (T367) is a suppressive phospho-event, our results are consistent with the idea that p38-dependent suppressive phosphorylation of EZH2 leads to diminished trimethylation of H3K27, thus favoring ER α expression.

The new data are added in the revised manuscript as:

- Figure 7 a-d.
- TEXT in the RESULTS section (the same sentence as in the Authors’ response R.1.1):

”In addition, consistent with our earlier notion suggesting involvement of p38 as a mediator of the matrix stiffness to EZH2-mediated trimethylation of H3K27 and downmodulation of ER α activity, inhibition of p38 also suppressed... the expression of ER α is favored.”

- DISCUSSION: The same sentence as in R1.1. (“ In support, we show that the inhibition of p38...”

We consider that these new data provide sufficient, although not yet comprehensive, evidence to support the epigenetic aspect in the ER α regulation by p38.

The relevant experiments and the results are detailed above in the Authors’ response to R1.1.

The other aspect is that some of the data presented are not novel and are correlative. The association between high EZH2 and negative ER expression has been reported in 2003 (Kleer et al), and later corroborated by several studies.

Authors' response (R.1.3): We wish to stress that our manuscript finds that ER α expression is regulated through matrix stiffness mediated activation of p38 in a highly physiological model of mammary tissue and breast cancer. We are aware of the negative association between EZH2 and ER α in breast cancer and cited Kleer et al. 2003 and Anwar et al. 2003 in our original manuscript. To further acknowledge the previous findings, we added a sentence to our revised manuscript (in RESULTS): "The negative correlation between EZH2 and ER α is also consistent with earlier studies, that describe increased levels of EZH2 in ER α negative breast cancer." We also cited one more recent paper showing a role for EZH2 in ER α - breast cancer (PMID: 31968251). These results are consistent with our findings, which have been moved in the revised manuscript to Supplementary Fig. 8c

The association shown between phospho p38 and ER in the 17 invasive carcinomas is preliminary. Additional cases are needed to conclude a statistically significant association in cancer.

Authors' response (R.1.4): To address the criticism, we analyzed 25 additional invasive breast cancer specimens for their phospho-p38 and ER α expression. Hence, the total number of tumors stained is now 42. The higher sample number allowed a statistical analysis for significance. According to the Pearson's product-moment correlation the correlation was 0.98 with a p-value of 0.025, and thus the correlation between phospho-p38 and ER α in invasive breast cancer is statistically significant. In the revised manuscript, we have added the statistics to Figure 7e and we show the raw data (IHC stainings) in Supplementary Figure 7.

Data in Fig. 7 is correlative. The association between mammographic density and ER expression is interesting. However, data are shown in normal tissues, and not in cancer. Further study in cancer is warranted.

Authors' response (R.1.5): Increased breast tissue stiffness is considered as one of the major risk factors for breast cancer and therefore, the mammographic screening is performed to detect the early onset of breast cancer and to find those at risk. We used the data pertaining to the latter situation. The tumor tissue detected from mammographic images exceeds all four stiffness categories (Grade 1-4) that the radiologists are using to grade the normal breast tissue. Therefore, no similar gradings or mammographic data exists for breast tumor tissue. However, our explant model of breast cancer strongly suggest that the tissue stiffness also regulates ER α expression in context of breast cancer.

Reviewer #2:

The present manuscript from Munne et al describes experiments that aim to establish an *ex vivo* luminal ER α + breast cancer model with patient-derived breast explants to restore ER α expression and study ER α + signalling pathway. Using MMECs and an impressive number of human breast tissue and breast tumor samples, the authors show that maintaining luminal epithelial phenotypes for both normal breast and breast cancer explants *ex vivo* is not cell identity-dependent, but matrix-dependent. They distinguish between matrix scaffolds which are luminal preserving matrices (LMx) and basal promoting matrices (BMx). The transcriptomic profiles and ER α expressions in MMEC are matrix stiffness-dependent and mediated by stress and H3K27me3 pathways. The authors demonstrate that about 20-fold higher effective stiffness is required to activate stress and hormonal pathways in the human explants than in the mouse explants. In summary, matrix stiffness induces p38 stress pathway, represses EZH2-dependent H3K27me3, and upregulates ER α expression. These findings if fully supported by data are novel and will be of interest to the field.

The main concern about this manuscript is whether the stiffness of agarose matches the stiffness of human breast carcinoma, which determines the biological/clinical relevance of this *ex vivo* culture model.

From many works of Valerie M. Weaver's, the human breast carcinoma has a stiffness of around 2 kPa measured by AFM, which is much lower than the effective elastic modulus (indicating stiffness) of agarose+compression (373kPa) used in this manuscript. Could authors explain where the difference of stiffness comes from? Is there a dependency with the techniques used (the authors used rheological test instead of AFM)?

Major Concerns:

1. The main concern about this manuscript is whether the stiffness of agarose matches the stiffness of human breast carcinoma, which determines the biological/clinical relevance of this *ex vivo* culture model. From many works of Valerie M. Weaver's, the human breast carcinoma has a stiffness of around 2 kPa measured by AFM, which is much lower than the effective elastic modulus (indicating stiffness) of agarose+compression (373kPa) used in this manuscript. Could authors explain where the difference of stiffness comes from? Is there a dependency with the techniques used (the authors used rheological test instead of AFM)? Acerbi, Irene, ..., Valerie M. Weaver. *Integrative Biology* (2015) The authors referred matrix stiffness to different terminologies, e.g. storage modulus, elastic modulus and complex modulus, which is difficult and confusing for readers without professional Materials background. Could authors find a way to make it simpler and clearer? For instance, would it work that the authors specify the relationship between each terminology and use one of them to represent stiffness consistently?

Authors' response (R.2.1): The reviewer makes an important remark by pointing out the apparent discrepancy in stiffness values earlier documented for human breast cancer (2-40 kPa; PMID: 25959051, PMID: 17327649) and agarose+compression model (373 kPa) used in the present study. As the reviewer anticipated, the heterogeneity of tissues and instrumental techniques affect the absolute stiffness values obtained for different materials. For example, we note a systematic study by Wu et al. (*Nat. Methods*, 2018, 15, 491–498: *A comparison of methods to assess cell mechanical properties*), which compared Atomic Force Microscopy (AFM), magnetic twisting cytometry, particle tracking microrheology, parallel plate rheometry, cell monolayer rheology and optical stretching of tumorigenic and non-tumorigenic cells. They found significant differences with values varying by several orders of magnitude from one technique to the other.

What comes to estimating breast cancer stiffness, several earlier studies have used Atomic Force Microscopy (AFM) technique, which is a highly relevant method when one estimates the micro-scale stiffness for a variety of different biological surfaces such as tissues and cells. However, the AFM method is not suited to estimate the elastic properties of bulk 3D matrices. We used rheometry, since the principal goal of our research was to obtain relative values to estimate the pressure caused by different matrices on cell cultures. In other words, we sought to find via head-to-head comparison suitable matrix/scaffolds with proper mechanical properties for retention of cellular identity, heterogeneity, and expression of hormone receptor of patient-derived breast cancer tissues. The metrics provided in this study should be considered only as a technical parameter that allows side-by-side comparisons of different matrix stiffnesses and our stiffness values cannot be compared with the stiffness values obtained via AFM or other similar techniques.

For general reader, we have clarified the use of rheology for the specific purpose of this study with the following sentence:

- TEXT in the RESULTS section:

“To define the relative stiffness of each matrix, we used rheological measurements to obtain the elastic modulus (stiffness) of each gel. We note that atomic force microscopy (AFM) is one widely used technique to evaluate stiffnesses from biological surfaces, but this method is not suited to estimate the elastic properties of larger gel volumes and the stiffness values obtained via AFM or other similar techniques cannot be directly cross-referenced with the metrics obtained via rheological measurements (Wu et al., *Nat. Methods*, 2018, 15, 491–498: *A comparison of methods to assess cell mechanical properties*). Therefore, the metrics provided in this study should be considered only as a technical parameter that allows side-by-side comparisons of different matrix stiffnesses and our stiffness values cannot be compared with the stiffness values obtained via AFM or other similar techniques .”

As requested by the reviewer, we have also provided the definition of the terminologies used in the supporting information. To help the interdisciplinary readers, we have provided a small glossary of the terms used in rheometric studies:

- TEXT in the SUPPORTING INFORMATION: Glossary of the rheometric terms

In short, we explain why we chose to use the rheometry and that the stiffness values obtained in rheometric analysis are not directly comparable to those obtained by Atomic Force Microscopy (AFM) techniques.

2. The authors showed that epithelial cell identity was not a stable feature in a culture but highly sensitive to changes mediated by the matrix environment, and they demonstrated this by comparing the stiffness differences mainly between alginate (soft) and agarose (stiff), as well as in agarose with different stiffness. However, are there other aspects, such as structural and/or chemical property differences among alginate, agarose and other luminal preserving materials tested (egg white and ovomucin), that needs to be considered? For instance, if the authors could acquire stiffness gradients in other materials than merely in agarose by changing the material concentration or applying the magnet-mediate compression that they used to further increase the stiffness of agarose, is it possible that ER expression and function would also be restored?

Authors' response (R.2.2): We explored all our luminal phenotype preserving matrices with SEM and rheology, finding that the ultrastructure of the materials and mechanical properties varied substantially (Figure 2a, Supplementary Figures 2, 3a and 3b). The bioinert nature of the matrices was the only clear

unifying parameter between the luminal preserving matrices, although we cannot rule out the possibility that some subtle materials-specific factors also influenced the stability of the luminal phenotype. Based on our findings, we tend to believe that any other bioinert polymer could in principle replace agarose and our magnet mediated compression system, if only the material provides sufficient compressive force. We tried to find such polymers (besides agarose), but the constraints come from the difficulties to embed cells inside the matrix when the polymer concentration increases. We tested several materials such as: chitosan, hyaluronan, different nanocelluloses, epoxy resin, gellan gum, carrageenan. We also tried to introduce different crosslinking strategies and additives such as hydroxyapatite or glass to some of these materials in order to increase the stiffness. Agarose was the only material, that enabled us to reach a high level of stiffness yet not damaging the cells. Additionally, it was the only material, that could retain its conformation under the magnetic mediated compression, unlike the other tested materials, which either broke or spread under the compression.

Minor Concerns:

3. In line 361: the authors claim a strong positive correlation between p38 and ER α protein expression levels. However, the R2 value is only 0.12, and the Spearman and Pearson correlation coefficient are around 0.3.

Authors' response (R.2.3): The reviewer correctly points out that the correlation is only weak between p38 protein level and ER α . Since p38 is a phosphorylation regulated protein, we performed an immunohistochemical staining to define the correlation between phospho-p38 and ER α in 42 invasive breast cancer samples. The correlation between phospho-p38 and ER α was statistically significant (Pearson: 0.98). The new data are now shown in Figure 7 e and in Supplementary Figure 7. We have moved the previous data from Figure 7 a, c, and d in the Supplementary Figure 8 and removed the statement of a strong positive correlation.

4. Have the authors tried to induce ER α expression by anisomycin with other TNBC cell lines than DU4475 cell line?

Authors' response (R.2.4): For the revision, we treated five new TNBC cell lines with anisomycin to test the induction of ER α expression. Three of these cell lines upregulated phospho-p38 and ER α in response to the anisomycin treatment (BT-20, HCC1806, HCC38), while two cell lines did not respond to the treatment (BT549, MDA-MB-468). Therefore, altogether four out of six tested TNBC cell lines upregulate ER α in response to anisomycin. We added the new data in the Supplementary Figure 5 f. We did not study further the reasons for the non-responses in BT-549 and MDA-MB-468 cell lines, but potential reasons could be that the signaling pathways required for p38-mediated ER α expression are mutated in these cell lines, or that ER α expression is suppressed in a different way.

5. The authors chemically induced stress by anisomycin and this successfully upregulated the ER protein expression in TNBC cell line. Have the authors tried to mechanically induce stress and ER α expression in TNBC cell lines or TNBC patient derived breast cancer explants by magnet-mediated compression?

Authors' response (R.2.5): Plastic cell culture in a standard dish exerts high stiffness and indeed, the compressive forces to the cells are higher than in tumor tissue (see Supplementary Figure 9 in the manuscript). We chose to use DU4475 cell line, since it grows in a suspension (not exposed to any compressive forces) unlike the other TNBC cell lines, which grow on a petri dish. Anecdotally, despite

the high stiffness caused by the 2D cell culturing, the TNBC cell lines remain ER α negative. We believe that the TNBC tissue remains ER α negative in petri dish and tumors due to certain mutations or other tumor-specific mechanisms that affect the pathways discovered in the present study and these mechanisms are part of our current studies.

6. Anisomycin is a potent activator of stress-activated protein kinases (JNK/SAPK) and p38 MAP kinase. Acts as a potent signaling agonist to selectively elicit homologous desensitization of immediate early gene induction (c-fos, fosB, c-jun, junB and junD). How they can rule out the inhibition of JNK as equally important for ER?

Authors' response (R.2.6): In the revised version we have addressed the role of JNK by using JNK inhibitor (SP600125) in a series of experiments. We show that in the conditions that preserve ER α (LMx-Ag in MMECs and magnet mediate compression in PDEC-BC) inhibition of JNK does not reduce ER α expression or activity, while inhibition of p38 activity clearly did so Figure 7a-d; Supplementary Figure 6 c.

To explore the role of JNK, we used a specific JNK inhibitor (SP600125) to define whether JNK inhibition affects the expression of ER α , p38p, EZH2 p(T367), and H3K27me3 in LMx-Ag cultured MMECs and magnet compressed PDEC-BCs. Our results show that, unlike p38 inhibition, JNK inhibition does not have any clear effect on ER α expression in the immunofluorescent stainings or western blot analysis, thus suggesting no involvement for JNK pathway in the stiffness mediated regulation of ER α .

The new data are added in the revised manuscript as:

- Figure 7 a-d; Supplementary Figure 6 c.
- TEXT in the RESULTS section:

We also explored the involvement of JNK, which is another key stress-activated protein kinase (SAPK), using a specific JNK inhibitor (SP600125). These experiments showed that, unlike p38 inhibition, JNK inhibition did not affect the stiff matrix dependent expression of ER α or change the p38/EZH2/H3K27me3 activity in LMx-Ag cultured MMECs or magnet compressed PDEC-BCs (Figure 7a-d). Also, no alteration in the ER α regulated gene sets were observed after JNK inhibition although JNK regulated gene sets were clearly suppressed (Supplementary Fig. 6c). Therefore, the experiments do not find a role for JNK in stiffness mediated regulation of ER α .

- DISCUSSION: The same sentence as in E.1 (" In support, we show that the inhibition of p38..."

7. In Methods, the authors did not specify the experimental process of tissue embedment in matrix in either "Isolation of Biological Material and Three- Dimensional (3D) Culturing" or "Preparation of 3D Matrices" section, is it possible for the authors to provide experiment procedures and details so that other researchers can reproduce the experiment?

Authors' response (R.2.7): As suggested by the reviewer, we have extended the description of the tissue processing methods in the Methods section under the subtitle: "Preparation of 3D Matrices". Additionally, for the revision we established a new method to extract proteins from the agarose gels (LMx-Ag), which is now specifically described in the Methods section under the subtitle: "Protein Extraction from LMx-Ag cultures."

8. The authors did not provide the images of CTRL conditions serving as references or base lines for comparison in some figures. For instance, in figure 2a and 2b, it would be nice to have immunofluorescence images of MMEC and PDEC-N that were not cultured in any of the matrices to provide naïve expression levels of luminal and basal markers. It is also helpful if authors specify the sample code in every figure, for example, in figure 1c, the authors only marked PDEC-BC with a sample code, but did not mark PDEC-N.

Authors' response (R.2.8): The non-cultured day 0 control samples of MMEC and PDEC that were not cultured in any of the matrices are shown in Figure 1 c. The figure shows the baseline for luminal (CK8) and basal (CK14) cytokeratin expression for MMEC, PDEC-N, and PDEC-BC. In the revised version we have added the following note in the figure legend:” For comparison to the day 0 sample, see Figure 1c.”

We hope that these changes make the relevant comparisons clearer. For consistency, we have added the sample code to the PDEC-N in Figure 1c. In the other figures we have added the sample code when we thought that adding the code is informative.

For the reviewer: The figure shows additional samples at the day 0 to represent the baseline for CK8 and CK14 in five additional PDEC-BC and one PDEC-N.

9. In figure 2a, the authors did not show images of tissue embedded in ovomucin even though they mentioned this in the figure legend and in the main text.

Authors' response (R.2.9): We thank the reviewer for pointing this out. The reason why we did not add a figure representing ovomucin was the space constrains, but the rheological measurements are presented. Ovomucin represents the gelling component of egg white, and the phenotype of the explant was identical to the egg white grown explants. Here is an example image (for the reviewer) of MMEC cultured in ovomucin for 7 day. The sample is stained for luminal CK8 (red) and basal CK14 (green) cytokeratin expression. Scale bar 10 μ m.

For revision, we have added the following notion in the figure legend: “The phenotype of ovomucin grown explant was identical to egg white grown explants.”

10. In line 266 “the ER α -regulated gene sets were clearly diminished in the treated samples (Fig. 4I)”, there is only one gene PGR in figure 4I, no gene sets. In figure S3e, there is another gene GREB1. The author need to either put two gene expression plots together, or mention the supplementary figure in the brackets at the end of this sentence.

Authors’ response (R.2.10): As suggested by the reviewer, the text has been corrected and the gene sets are shown in the Figure 4k. We also now mention both PGR and GREB1 in the text.

11. In line 286, the authors mentioned the PDEC-N was shown in figure 4 and figure S5, but there is no enrichment map of PDEC-N in figure S5.

Authors’ response (R.2.11): We thank the reviewer for pointing out this mistake. It has been corrected in the revised version.

12. In figure S2, the author did not specify the difference between figure S2 m and S2 n with egg white.

Authors’ response (R.2.12): We thank reviewer for pointing out this unclarity. The two egg white images in Supplemental Figure 2 m & n shows the structural variation within the egg white gel. We added a text “field of view” (FoV) to the figure and added a following sentence: “Egg white, two different fields of view (FoV) are presented to show the structural variation within a gel”.

13. Lines 88-90: authors do not mention PDXs. There are few other models that can retain functional ER α signaling (e.g PMID: 26947176).

Authors’ response (R.2.13): We agree that citation to the work of George Sflomos and Cathrin Brisken is appropriate in the context of the paper. We have added the following text in the introduction: “In vivo, stable ER α expression has been reported in patient-derived xenograft (PDX) models, especially in tumor cells introduced via intraductal transplantation^{8,9} and these findings have suggested a strong microenvironment-dependent dynamic component in the regulation of ER α expression.”

14. Line 103-104. Authors should mention recent study of 2% alginate (PMID:32807212).

Authors’ response (R.2.14): We thank reviewer for notifying us this recent publication. A citation has been added.

15. Make sure the contexts *in vivo* and *ex vivo* are all in Italic in the paper.

Authors’ response (R.2.15): We thank the reviewer for pointing out the formats. The text has been now italicized.

16. In the legend of Fig 2, there is a typo in the function $E = 2(1-v)G^*$. The correct one should be $E = 2(1+v)G^*$.

Authors’ response (R.2.16): We thank the reviewer for pointing out the typo. This has been corrected.

17. Typo in line 661: there are two “where”, delete one.

Authors’ response (R.2.17): The typo has been corrected.

18. Typo in legend of figure S3f, 24 may be 24 h.

Authors’ response (R.2.18): The typo has been corrected.

19. Fig. 4i: to write ER α and not ERa

Authors’ response (R.2.19): The typo has been corrected.

20. Authors they should report more comprehensively in the methods how to construct agarose extracellular matrix scaffold.

Authors’ response (R.2.20): We have now extended the protocol for agarose scaffold in Methods section: “Agarose solutions were prepared by first dispersing 0.07g of the UltraPure™ Low Melting Point Agarose (Invitrogen, 16520050) in 1ml of 1xPBS followed by heating the mixtures ... 0.5 ml of Mammocult media per 8-chamber slide well was added after the matrix was solidified.”

Reviewer #3

In this study, Munne et al analyze the regulation of estrogen receptor (ER α) in normal and transformed mammary epithelial cells. They use 3D cultures under distinct conditions to demonstrate that matrix density and stiffness, stress signaling (p38) and reduced EZH2 mediated histone methylation are all important to sustain ER α expression in these cells. Notably, using human breast tissue from mammaplasty, they show a correlation between active p38 stress signaling and ER α expression. Moreover, they show that breast tissue density was associate with increased ER α expression. There are several interesting aspects to this study. It proposes microenvironmental regulation of ER α expression in mammary cells and this may be highly consequential. However, there are a number of important caveats and loose ends that need attention. The study is rather fragmented and therefore it is difficult to get a clear overall picture of what is going on. Details below.

Major points:

1. The authors demonstrate that tissue density and stiffness, stress signaling and EZH2 inhibition all can promote ER α expression. However, the link between the three functions and which stress signaling pathway is involved are not demonstrated clearly and thus still rather speculative. For example, what is shown in Figure 5d is not sufficient to demonstrate a link between tissue stiffness and p38 signaling. First, the comparisons are confusing. Why are the same comparisons not used for MMEC and the PDECs?

Authors’ response (R.3.1.1): In the revised manuscript we provide new data to functionally link the matrix stiffness, stress signaling dependency, EZH2 and ER α expression (Figures Figure 7 a-d; Supplementary Fig. 6 a, b, d) and we show new data dissecting better the involvement of p38 and not

JNK pathway (Figure 7 a-d; Supplementary Figure 6 c). We show new evidence in our revised manuscript to support the role of p38p in matrix stiffness mediated activation of ER α by using p38 inhibitors.

In the revision, we have performed additional experiments with two different p38 MAPK inhibitors (RWJ67657, SB203580) and the new results consistently show that inhibition of p38 activity suppresses ER α expression and activity in the stiff matrix conditions (MMEC in LMx-Ag, PDEC-BC in the magnet compressed LMx-Ag).

The new data are added in the revised manuscript as:

- FIGURES: Figure 7 a-d; Supplementary Fig. 6 a, b, d
- TEXT in the RESULTS section:

”To test the functional importance of p38 mediated stress pathway for ER α expression, we chemically inhibited p38 in LMx-Ag cultured MMECs and magnet compressed PDEC-BCs (for validation of p38 MAPK inhibitors, see supplementary Fig. 6 a, b, d). As evidenced by the western blot, RNA sequencing, and immunofluorescence microscopy analysis, inhibition of p38 abolished nuclear ER α expression (Figure 7 a-d) and suppressed ER α activity in both MMEC and PDEC-BC (Supplementary Fig. 6 b, d).”

Furthermore, the inhibition of p38 in the same experiments also suppressed phosphorylation of EZH2-p(T367) and resulted in enhanced trimethylation of H3K27

The new data are added in the revised manuscript as:

- FIGURES: Figure 7 b, d
- TEXT in the RESULTS section:

”In addition, consistent with our earlier notion suggesting involvement of p38 as a mediator of the matrix stiffness to EZH2-mediated trimethylation of H3K27 and downmodulation of ER α activity, inhibition of p38 also suppressed phosphorylation of EZH2-p(T367) and resulted in enhanced trimethylation of H3K27 (Figure 7 b, d). Since phosphorylation of EZH2 at T367 suppresses its activity, our results from both mouse and human explant cultures altogether are consistent with a mechanistic model presented in Fig 6 e. Accordingly, specifically a stiff matrix induces a p38 mediated stress pathway, which keeps EZH2 phosphorylated at T367 thus suppressing the activity of this key enzyme that catalyzes the addition of methyl groups to histone H3 at lysine 27. In the absence of epigenetic repression, the expression of ER α is favored.”

- DISCUSSION:

”In support, we show that the inhibition of p38 simultaneously prevented the phosphorylation of EZH2 and led to an increase in methylation of H3K27 while ER α was suppressed. This effect was specific to p38 as inhibition of JNK did not have similar effects on ER α expression.”

The new results clearly demonstrate a requirement for p38 in regulation of H3K27me3 and ER α in the mouse and human mammary epithelial cultures and in breast cancer samples.

It is of note that for the new experiments, we needed to establish a completely new method for extracting proteins from agarose gels, which is now described in the:

- METHODS section: “Protein Extraction from LMx-Ag cultures”.

In the revised manuscript, we show with western blot analysis that in stiff matrices, inhibition of p38 suppresses EZH2-p(T367) in both the MMEC and PDEC-BC cultures. Since phosphorylation of EZH2 at (T367) is a suppressive phospho-event, our results are consistent with the idea that p38-dependent suppressive phosphorylation of EZH2 leads to diminished trimethylation of H3K27, thus favoring ER α expression.

The new data are added in the revised manuscript as:

- Figure 7 a-d.
- TEXT in the RESULTS section (the same sentence as above):

”In addition, consistent with our earlier notion suggesting involvement of p38 as a mediator of the matrix stiffness to EZH2-mediated trimethylation of H3K27 and downmodulation of ER α activity, inhibition of p38 also suppressed... the expression of ER α is favored.”

- DISCUSSION: The same sentence as above (“ In support, we show that the inhibition of p38...”

The reason for not showing the same comparisons for MMECs and PDECs in the Figure 5d was that without magnet mediated compression, a sufficient matrix stiffness was not achieved for PDECs. We solved this problem later in the manuscript by inventing the magnetic mediated compression system, which we show in the Figure 6. For the revised manuscript, we have added a comparison of PDEC-BCs grown either in the non-ER α supporting LMx-Ag or the ER α -supporting magnet compressed LMx-Ag (Supplemental Figure 5 e). The results show enrichment of ER α and p38p mediated signaling in the magnet compressed LMx-Ag as compared to the non-compressed LMx-Ag (three different patient samples).

What type of stress signature is used? Is it specific p38 induced signature or does it include JNK induced genes? This should be clarified.

Authors’ response (R.3.1.2): We thank the reviewer for pointing out this potential issue and, the stress signature we used indeed comprises both p38- and JNK-induced genes. To dissect the roles of p38 and JNK, we investigated the role of JNK using specific JNK inhibitor (SP600125) and explored whether the JNK inhibition would also affect the expression of ER α , p38p, EZH2 p(T367), and H3K27me3 in LMx-Ag cultured MMECs and magnet compressed PDEC-BCs. Our results clearly show that, unlike inhibition of p38, JNK inhibition does not have effect on ER α expression based on immunofluorescent staining and western blot analysis. Furthermore, no alteration in the ER α regulated gene sets were observed after JNK inhibition, when JNK regulated gene sets were suppressed demonstrating efficacy of the JNK inhibitor. Therefore, the new data clearly demonstrates that the stiffness mediated regulation of ER α depends on p38 and not JNK.

The new data are added in the revised manuscript as:

- Figure 7 a-d; Supplementary Figure 6 c.
- TEXT in the RESULTS section:

“We also explored the involvement of JNK, which is another key stress-activated protein kinase (SAPK), using a specific JNK inhibitor (SP600125). These experiments showed that, unlike p38 inhibition, JNK inhibition did not affect the stiff matrix dependent expression of ER α or change the p38/EZH2/H3K27me3 activity in LMx-Ag cultured MMECs or magnet compressed PDEC-BCs (Figure 7a-d). Also, no alteration in the ER α regulated gene sets were observed after JNK

inhibition although JNK regulated gene sets were clearly suppressed (Supplementary Fig. 6c). Therefore, the experiments do not find a role for JNK in stiffness mediated regulation of ER α .”

- DISCUSSION: The same sentence as above (“ In support, we show that the inhibition of p38...”

Moreover, to confirm a link between ECM stiffness and p38 signaling, a p-p38 Western blot should be performed on cells in matrices of different stiffness.

Authors’ response (R.3.1.3): In the revision, we demonstrate the positive effect of increasing agarose matrix stiffness on p-p38 expression in the MMECs. The new data are included in the Supplementary Figure 3g.

It is also important to note that anisomycin, that the investigators use to promote stress signaling is not specific for p38 but can also induce JNK signaling. Which stress pathway is required for ER α expression. To sort this out, ER α expression should be analyzed in MMEC grown in LMx-Ag and treated with JNK or p38 inhibitors.

Authors’ response (R.3.1.4): The new findings in the revised manuscript are consistent with the notion that p38 is the major mediator between the matrix stiffness and ER α regulation. To explore the role of JNK, we used a specific JNK inhibitor (SP600125) to define whether JNK inhibition affects the expression of ER α , p38p, EZH2 p(T367), and H3K27me3 in LMx-Ag cultured MMECs and magnet compressed PDEC-BCs. Our results show that, unlike p38 inhibition, JNK inhibition does not have any clear effect on ER α expression in the immunofluorescent stainings or western blot analysis, thus suggesting no involvement for JNK pathway in the stiffness mediated regulation of ER α .

The new data are added in the revised manuscript as mentioned the Authors’ response to R.3.1.2.

To confirm the connection of stiff matrix and stress to EZH2 repression, T367 phosphorylation of EZH2 should be analyzed under the same conditions (LMx-Ag with stress signaling inhibitors). These experiments could also be done with PDEC-N and PDEC-BC in compressed LMx-Ag matrix and with the stress signaling inhibitors.

Authors’ response (R.3.1.5): For the revised manuscript, we have added new data to demonstrate that p38p inhibition reduces EZH2-p(T367) in the stiff matrix both in the MMEC and PDEC-BC cultures. Since phosphorylation of EZH2 at (T367) is a suppressive phospho-event, our results are consistent with the idea that p38 dependent suppressive phosphorylation of EZH2 prevents trimethylation of H3K27, thus enabling ER α expression.

We addressed this comment by testing the effect of p38 inhibition on matrix stiffness mediated phosphorylation of EZH2-p(T367) in LMx-Ag cultured MMECs and magnet compressed PDEC-BCs. In the revised manuscript, we show with western blot analysis that in stiff matrices, inhibition of p38 suppresses EZH2-p(T367) in both the MMEC and PDEC-BC cultures.

The new data are added in the revised manuscript as mentioned in the Authors’ response to R.3.1.1.

We also performed a similar experiment in PDEC-N cultures that shows the suppression of phospho-38 and pEZH2 (T367) upon treatment with a p38, but not a JNK inhibitor. The data are shown below as figure for the reviewer.

2. I have several concerns about the phenotypic consequences of ER α activity. The data suggest that there may be a discordance between ER α and luminal phenotype (based on CK8 expression). Whereas LMx-AI, LMx-Ew and LMx-Ag all promote expression of CK8, only LMx-Ag promotes induction of ER α in MMEC and none of the matrices induce ER α in human PDEC samples. Would ER α signaling not be a key regulator of luminal fate in the mammary gland? This needs to be analyzed and explained much better.

Authors' response (R.3.2.1): Thank you for bringing this very important aspect of mammary biology up. It is generally thought that the CK8+ luminal phenotype as such does not require ER α expression since both luminal progenitor cells (ER α -) and mature luminal cells (ER α +) express CK8 (Shehata et al. PMID: 23088371). The pluripotent ER α - CK8+ cells have a capacity to differentiate into multiple different lineages (PMID: 20804960). Our data from *ex vivo* cultures suggest that the conversion of ER α -CK8+ to ER α +CK8+ phenotype involves microenvironmental cues, most importantly matrix stiffness. In the soft, luminal phenotype preserving matrices (LMx-Ew or LMx-AI), both the MMECs and PDECs expressed methylation and pluripotency related gene sets (Supplementary Figure 3 h-k), which proposes that the soft microenvironment may favor the ER α - progenitor cell phenotype. Only if the matrix stiffness was substantially increased, the ER α and ER α regulated gene sets became activated while the methylation and pluripotency signatures simultaneously disappeared (Figure 4 b vs. f). At this point, our data strongly suggest a key role for matrix stiffness in the ER α regulation (also in the physiological breast environment, Figure 7f), however, it still remains speculative how the matrix stiffness promotes the differentiation of luminal progenitor cells towards mature ER α + status during the mammary gland development. The stiffness signal may come for example from matrix crosslinking processes, stromal cells, myoepithelial cells, or any other possible mechanisms not studied here. While our results hopefully stimulate this type of studies, we believe that the mammary gland cell lineage focused studies are outside of the scope of the present study.

"none of the matrices induce ER α in human PDEC samples"

We wish to point out that the key findings are consistent between MMECs and PDECs despite of the difference in the stiffness magnitude. In the human PDECs, higher compressive forces were required for ER α expression (Figure 6). This may reflect the fact that human breast is naturally stiffer environment than the mouse fat pad, Supplementary Figure 9.

In the revision, the following new TEXT has been added to DISCUSSION: "Interestingly, during the mammary gland development, the pluripotent ER α - CK8+ cells have a capacity to differentiate into multiple different lineages, including differentiated ER α + CK8+ cells (PMID: 20804960) and there are multiple factors affecting matrix stiffness *in vivo*, such as extracellular matrix composition, fibroblasts or myoepithelial cells. It remains a challenge for future studies to define whether the developmentally regulated processes involve compressive forces to regulate ER α expression."

First, by using luminal gene signature rather than a single marker. In figure 3, only gene signatures that are induced in basal cells are used when comparing BMx and LMx conditions. How proficient are the LMx matrices in promoting luminal phenotype and is there a difference between LMx-Ag and the others? This should be addressed using specific luminal signatures.

Authors' response (R.3.2.2): Thank you for this question, we try to clarify. The Gene Set Enrichment Analysis (GSEA) is based on a collection of priori defined set of genes corresponding to the known molecular signatures or published data. We used those molecular signatures that were available in MSigDB/GSEA to distinguish between the basal and luminal phenotype, which included HUPER_BREAST_BASAL_VS_LUMINAL_DN/UP gene set, containing 53 and 58 genes that were up- or down-regulated, respectively, in the basal mammary epithelial cells compared to the luminal cells. In addition, we also used Δ Np63 gene set, since this reflects the master regulator switch in the basal differentiation (PMID: 10227293; PMID: 10594758; PMID: 10227294; PMID: 20379195). The enrichment of these gene sets were used to distinguished basal vs. luminal phenotype between two different culture conditions. The MSigDB data base does not contain any luminal enriched data sets. We also used specific cytokeratin markers that is a well-established approach to define the luminal and basal cell identity. The cells were luminal after the extraction (Figure 1c.) and the luminal matrices preserved the luminal phenotype of the cells, because they maintained the cell phenotype like it is right after the extraction. We believe that we analysed the basal vs luminal phenotypes as comprehensively as possible with widely accepted approaches or at least to an extent that should justify the main conclusions in the manuscript.

3. In Figure 6e, the investigators propose a model where ER α is a direct target of p38 signaling via EZH2. However, they only show that ER α protein is affected by stress signaling. ER α stability is tightly regulated and increased stability can be observed in breast cancer cells. Is the ER α mRNA changed?

Authors' response (R.3.3.1): The reviewer requested more clear evidence to couple the matrix stiffness-p38-EZH2 axis to ER α expression. For the revision, we provide extensive new data to support our claims. Please see our response at Authors' response to R.3.1.1.

With regards to *Esr1* mRNA expression, below we show the mRNA expression of *Esr1* gene in two PDEC-BC samples (P182T and P184T) from three different conditions. *Esr1* expression is

downregulated in soft ER α - LMx-Ew and BMx-Mat gels as compared to the ER α + uncultured samples. The expression of *Esr1* is normalized to GAPDH. The data are in line with our hypothesis of methylation-mediated silencing of *Esr1* in soft matrices. We are aware of the limitations of observing only protein or mRNA levels. Therefore, we used gene sets to indicate ER α activity whenever possible.

For the reviewer only: The figure demonstrates the difference in the *Esr1* gene expression in PDEC-BCs corresponding to two different patients.

4. Figure 5e shows that the luminal breast cancer cells MCF7 and T47D have active p38 signaling. How do they maintain ER α expression under normal conditions and without a specific stiff matrix?

Authors' response (R.3.4.1): The reviewer points to normal conditions where the MCF7 and T47D cells grow on a Petri dish. In the Supplementary Figure 9, we review the stiffness of various culture conditions based on published data and culturing cells on a 2D cell culture plastic exposes cells to even greater stiffness than they experience in the tumor. Thus, the plastic culture mediated stiffness may be involved in the maintenance of p38-dependent expression of ER α in MCF7 and T47D cells. However, it is also true that most cells lose ER α expression in culture conditions and it is still unclear how this exactly happens and these mechanisms are currently studied in our lab.

Would the results suggest that luminal breast cancer have higher p38 signaling compared to basal breast cancer? This should be looked at.

Authors' response (R.3.4.2): This is an interesting question and for thus we searched the literature for possible answers. Indeed, there are data from several studies suggesting that p-p38 is significantly higher in ER α + tumors (PMID: 23900300, PMID: 30352570, PMID: 27386378). However, the basal tumors are also stiff and our data from IHCs and western blots samples suggest that p-p38 is also expressed in TNBC samples (unpubl.). The reason for the loss of p38 mediated regulation of ER α could be related to specific mutations or other activated pathways in these cases. Thus, the lack of ER α in stiff tumours or in the presence of p38p remains an interesting research topic, which is currently being investigated in the lab.

5. An interesting aspect of the study is that it shows ER α regulation that is highly context dependent. This may have impact on cellular phenotypes and possibly breast cancer subtypes. Specification of basal

like breast cancer has been shown to be regulated by the microenvironment (Roswall P et al Nat Med 2018). A discussion of this should be included.

Authors' response (R.3.5.1): We thank the reviewer for pointing out this interesting *in vivo* connection and we have added a note to discussion in the revised version of the manuscript: “Interestingly, earlier studies have demonstrated a microenvironment-dependent conversion of basal-like breast cancer phenotype to ER α + phenotype *in vivo* via carcinoma associated fibroblast-dependent mechanism, although the role of matrix stiffness in this context still remains to be clarified (Roswall P et al. 2018)”.

6. In Figure 4j, Ki67% is shown to decrease in response to E2 treatment, even though ER α targets are upregulated with the same treatment. This is surprising and should be explained.

Authors' response (R.3.6.1): Estradiol is a multifunctional factor for normal mammary epithelium, which regulates both proliferation and glandular morphogenesis. These effects involve both proliferation promoting and antiproliferative (differentiating) estradiol actions. We believe that addition of high levels of estradiol exerts some antiproliferative action in normal mammary epithelial tissue (as the reviewer points out with regards to Figure 4 j that uses MMECs) and while we have not studied these antiproliferative mechanisms, they could be mediated for example by Tp53 as suggested in previous studies (PMID: 18556351, PMID: 9285694, PMID: 6366574). Indeed, we observed in some of our gene expression profiling that estradiol treatment activated *Trp53* expression profiles. However, while this link between estradiol and Tp53 in MMECs offers interesting avenue to follow in the future studies, we feel that further discussing this notion is slightly out of the scope in the current manuscript.

7. In Figure 4i, quantified ER levels are shown with a scatter plot. However, only one condition (LMx- Ag 70mg/ml) is shown. The results from all polymer concentrations should be included in the scatter plot.

Authors' response (R.3.7.1): As suggested by the reviewer, we have now added the quantifications of the ER α levels also of the other matrix concentrations (10-40 mg/ml) into the scatter plot and replaced Figure 4i with this new one.

8. I think that correlation analyses, that include mRNA or total protein levels of p38, MAP3K1 or other proteins regulated by phosphorylation, do have a rather limited value.

Authors' response (R.3.8.1): We agree with the reviewer. The evidence for the correlation of phosphorylated p38 and ERa has now been strengthened with IHC analysis of 42 invasive breast cancer samples (Figure 7f, Supplementary Figure 7). The higher sample number allowed a statistical analysis for significance. According to the Pearson's product-moment correlation the correlation was 0.98 with a p-value of 0.025, and thus the correlation between phospho-p38 and ERa in invasive breast cancer is statistically significant.

The analyses showing the correlations between mRNA or total protein levels of p38, Esr1, Pgr, EZH2, and MAP3K1 are now moved to the Supplementary Figure 8.

We have added the statistics to the Figure 7 f and added all new the IHC images in the Supplementary Figure 7.

Minor points:

1. In figure 3h, MCF7 does not cluster with the others, even under LMx conditions. Why not?

Authors' response (R.3.1): We note that MCF7 cells were grown on a ultra-stiff Petri dish conditions before placing them into different matrices (Supplementary Figure 9). In our experiments, we observed that MCF7 cells did not cluster with the tumor samples in soft gels (LMx-Ew and BMx-Mat), but they did in stiff gels (LMx-Ag). The most likely, yet highly speculative explanation is that a certain level of matrix stiffness is required to induce reprogramming of MCF7 cell transcriptomes.

2. Figure legends need more details to explain each panel. There is no need to explain the results, only what each panel contains (how samples were treated and analyzed and to clarify abbreviations).

Authors' response (R.3.2): As suggested by the reviewer, we have added details to the figure legends throughout the manuscript.

3. Which post-hoc tests were used with ANOVA in multiple comparisons? This should be described in figure legends or in methods.

Authors' response (R.3.3): Dunnett's multiple comparisons post hoc test was used with ANOVA. We added the missing information to the figure legends.

4. In Figure 6g, it is difficult to see p-p38 induction in the PDEC-BC sample. Is this correct?

Authors' response (R.3.4): We apologize for the originally sub-optimal quality of the Figure 6g. The p-p38 antibody gives quite weak signal, but we consistently observed this signal in stiff matrices. See three different patient samples stained with p-p38 antibody below (for reviewer only). For the revision we have replaced the original figure with a representative figure, which more clearly shows the p-p38 induction.

5. There are some minor issues with the Supplementary Figures. Suppl. Figure 3g, and Suppl. Figure 4 are not referred to in the main text. In addition, Suppl. Figure 8 is only one panel, but it is in the text referred to as Suppl. Figure 8a and 8b.

Authors' response (R.3.5):

We thank the reviewer for pointing out the missing citations and inconsistencies in the figures. The references to the figures have been added: Suppl. Figure 3g; Suppl. Figure 4 (Figure legend 3.). Suppl. Figure 9 a and b are corrected.

Reviewers' Comments:

Reviewer #2:

Remarks to the Author:

The concerns I had about the manuscript were adequately addressed by the authors.

Reviewer #3:

Remarks to the Author:

The authors have made substantial and thorough revisions to the manuscript. My concerns have been addressed adequately.

Reviewer #4:

Remarks to the Author:

The authors have added a significant amount of data to answer Reviewer #1's questions, and I think most of them have been sufficiently addressed. However, regarding Reviewer #1's major concern about whether inhibition p38 phosphorylation prevents the effects of matrix stiffness on H3K27me3 and on ERa expression, I have some suggestions and questions:

1. Even though the authors claimed that they couldn't harvest enough cells to do ChIP assays in order demonstrate H3K27me3 binding at ERa promoter, I wonder if they will be able to see H3K27me3 peak signals in the published ChIP-seq data that were generated in ERa+ breast cancer cells. This can potentially be supporting evidence that ERa is indeed epigenetically silenced by H3K27me3.
2. It would be the best if the authors can demonstrate by rescue experiment that EZH2 inhibition can reverse the effects of p38 inhibition on ERa expression and matrix stiffness.
3. In the original report about p38-mediated phosphorylation of EZH2, this post-translational modification promotes the cytoplasmic localization of EZH2 protein and promotes the association of cytosolic EZH2 with cytoskeletal regulators. Therefore, phosphorylation of EZH2 at T367 represents a PRC2-independent, non-canonical mechanism of EZH2 oncogenic function. I wonder if the authors saw the changes in the cellular localization of EZH2 upon p38 inhibition and if they can speculate how H3K27me3 is increased then.

Reviewer #4.

The authors have added a significant amount of data to answer Reviewer #1's questions, and I think most of them have been sufficiently addressed. However, regarding Reviewer#1's major concern about whether inhibition p38 phosphorylation prevents the effects of matrix stiffness on H3K27me3 and on ER α expression, I have some suggestions and questions:

1. Even though the authors claimed that they couldn't harvest enough cells to do ChIP assays in order demonstrate H3K27me3 binding at ER α promoter, I wonder if they will be able to see H3K27me3 peak signals in the published ChIP-seq data that were generated in ER α + breast cancer cells. This can potentially be supporting evidence that ER α is indeed epigenetically silenced by H3K27me3.

Authors' response (R.4.1): We thank the reviewer for providing us new ideas how to strengthen our claims in the manuscript. In our model, we would expect that ER α + cell lines show diminished H3K27me3 at *ESR1* promoter. We surveyed the published data on epigenetic modifications at *ESR1* promoter by using the Cistrome database (<http://cistrome.org/>), which collects the published ChIP-seq data from various cell lines and tissues targeting a wide range of DNA-binding proteins. The database has ChIP-seq data of the repressive H3K27me3 histone mark from different cell lines including breast cancer cell lines.

We found adequate datasets from 4 out of 7 TNBC cell lines and 5 out of 6 ER α + cell lines. The data from these four TNBC cell lines and five ER α + cell lines were included in the analysis and the reasons why 4 cell lines were omitted from the analysis are provided below. The data quality check and choice of cell lines was made by our bioinformatician, except for HCC1937 that we believe is not completely ER-negative cell line. We observed prominent H3K27me3 peaks in the *ESR1* promoters of all TNBC cell lines (MDA-MB-231, MDA-MB-436, MDA-MB-453, SUM159PT), while similar peaks were not observed in the five analyzed ER α + cell lines (MCF7, T47D, UACC812, ZR-75-1, ZR-75-30). These data demonstrate that *ESR1* promoter has H3K27me3 marks in TNBC and less so in the ER α + breast cancer cell lines. The reason for excluding 3 TNBC cell lines and one ER+ cell line from the analysis is described below.

ER α + Cell Line Datasets:

21NT: There are 2 H3K27me3 ChIP-seq datasets from this cell line. One dataset is from a heterofusion cell culture of SUM159 and 21NT cell lines and the other has low sequencing quality (UCSC Genome Browser Track image shown below)

TNBC Cell Line Datasets:

CAL51: There are 2 H3K27me3 ChIP-seq datasets from this cell line. One dataset is from a BAP1 knockout cell line and the other is from a MLL3 knockout cell line.

HCC1806: There are 2 H3K27me3 ChIP-seq datasets from this cell line. Both of them have low sequencing quality (UCSC Genome Browser Track image shown below).

HCC1937: This cell line is reported to be TNBC but it expresses ER α and is responsive to antihormonal therapies (our unpublished data).

Cell Line	Receptor Status	Dataset Usability
CAL51	TNBC	No available datasets from wild-type CAL51 cells
HCC1806	TNBC	No available datasets with high quality
HCC1937	TNBC	Reported as TNBC but shows ER expression
MDA-MB-231	TNBC	Dataset Used
MDA-MB-436	TNBC	Dataset Used
MDA-MB-453	TNBC	Dataset Used
SUM159PT	TNBC	Dataset Used
21NT	ER +	No available datasets with high quality or from wild-type 21NT cells
MCF7	ER +	Dataset Used
T47D	ER +	Dataset Used
UACC812	ER +	Dataset Used
ZR-75-1	ER +	Dataset Used
ZR-75-30	ER +	Dataset Used

The data shown below (Figure 1A) were added to the revision as Supplementary Figure 8e. The S8e has been cited in the text as: “ Intriguingly, our survey of the published data on epigenetic modifications at *Esr1* promoter... – further pointing to epigenetic mechanisms in downregulation of ER α .”

Out of interest, EZH2 that we have studied in the context of H3K27 trimethylation, also regulates DNA methylation in EZH2-target promoters (PMID: 16357870). Using DepMap database, we observed an overall increase in the DNA methylation at *ESR1* locus in the TNBC cell lines as compared to ER α + cell lines (Figure 1B), which further suggests a role for EZH2 mediated epigenetic regulation in *ESR1* expression. We show these data to the Reviewer only (Figure 1B).

These new data are all consistent with the role of H3K27 trimethylation in regulation of *Esr1* expression although other silencing mechanisms may also play a role.

A

Figure 1A) The figure shows histone methylation peaks at the promoter region of *Esr1* in TNBC (MDA-MB-231, MDA-MB-436, MDA-MB-453, SUM159PT) compared to ER α + (MCF7, T47D, UACC812, ZR-75-1, ZR-75-30). Data from the ChIP-seq of the repressive H3K27me3 histone mark. The peaks are visualized using Integrative Genomics Viewer (IGV). B) The figure shows expression and methylation of *Esr1* gene in TNBC breast cancer cell lines compared to ER α + cell lines. * $p < 0.05$, *** $p < 0.0001$ with Mann-Whitney U test.

B

2. It would be the best if the authors can

demonstrate by rescue experiment that EZH2 inhibition can reverse the effects of p38 inhibition on ER α expression and matrix stiffness.

Authors' response (R.4.2): We thank the reviewer for this idea of performing a rescue experiment with EZH2 inhibitor to further strengthen our hypothesis suggesting EZH2 as a key negative regulator of stress-induced ER α expression. Earlier, we showed that in the soft matrix (absence of p38 activation) the EZH2 inhibitor GSK-126 rescues the nuclear ER α expression in PDEC-BC, PDEC-N, MMECs, and DU4475. For the current 2nd revision, we repeated the experiment in stiff matrix by using the magnet compression method on PDEC-BCs. We added the p38 inhibitor (RWJ67657) to the cultures with or without EZH2 inhibitor (GSK-126), finding that that if p38 was inhibited, then H3K27 trimethylation was increased (Fig 2A). This p38 inhibition-dependent increase in the H3K27 trimethylation could be prevented (rescued) by the EZH2 inhibitor. Consistent with the involvement of EZH2 in the suppression of ER α expression, also the ER α expression was downregulated by the p38 inhibition, and this downregulation was rescued with the EZH2 inhibitor (Fig 2B).

These new data are fully consistent with our hypothesis, suggesting that the p38 mediated stress signaling in the stiff matrix environment maintains ER α expression by antagonizing the EZH2 mediated epigenetic suppression of ER α expression (mechanistically at least partly via H3K27 trimethylation of *Esr1*).

The data shown below (Figure 2A-B) were added to the revision as Figure 7e. The Fig. 7e has been cited in the text as: “ When we applied p38 and EZH2 inhibitor together ... the expression of ER α is favored whereas in the presence of EZH2 activity (H3K27me3 high), the expression of ER α is downmodulated.

Figure 2A) Western blot shows the effect of p38 inhibitor (RWJ67657) with and without EZH2 inhibitor (GSK-126) on H3K27me3. B) Immunofluorescent images of magnet compressed PDEC-BCs treated with and without p38 inhibitor (RWJ67657) and EZH2 inhibitor (GSK-126). Expression of ER α is shown in green and F-actin is shown in red. Scale bar 10 μ m.

3. In the original report about p38-mediated phosphorylation of EZH2, this post-translational modification promotes the cytoplasmic localization of EZH2 protein and promotes the association of cytosolic EZH2 with cytoskeletal regulators. Therefore, phosphorylation of EZH2 at T367 represents a PRC2-independent, non-canonical mechanism of EZH2 oncogenic function. I wonder if the authors saw the changes in the cellular localization of EZH2 upon p38 inhibition and if they can speculate how H3K27me3 is increased then.

Authors' response (R.4.3): The commercially available antibody for EZH2 p(T367) from Invitrogen/ThermoFisher Scientific (PA5-106225), which we have used throughout our study has been reported to work only in the western blot analysis. We tested whether we could use this antibody for IF-staining and the figure below shows the EZH2 p(T367) staining in the magnet compressed PDEC-BC samples, with and without a p38 inhibitor. It appears that the EZH2 p(T367) phosphosignal disappears in response to p38 inhibition (Figure 3A-B). This would be consistent with the idea that p38 phosphorylates EZH2 p(T367) under magnet compressed conditions (since without p38 the EZH2 phosphosignal disappears). However, we do not believe that we can really say much about the subcellular localization or mechanisms with the poor resolution obtained with the PA5-106225 antibody. Therefore, the data are presented for reviewer only. However, **we have added the following clarification to the 2nd revision: “While our current studies did not explore the exact biochemical mechanisms how p38 mediated phosphorylation of EZH2 impacts H3K27 trimethylation at *Esr1* locus, we wish to point out earlier studies,...”**

Figure 3A) Immunofluorescent staining of magnet compressed PDEC-BCs with and without p38 inhibitor (GSK-126). Expression EZH2 p(T367) shown in white. Scale bar 10 μ m. B) Western blot shows the inhibition of the EZH2 p(T367) after the p38 inhibition.

Reviewers' Comments:

Reviewer #4:

Remarks to the Author:

The authors have added a sufficient amount of data, and I don't have any further questions for them.